# Revisiting CroPA: A Reproducibility Study and Enhancements for Cross-Prompt Adversarial Transferability in Vision-Language Models

**Atharv Mittal**[*]                                                     *atharv_m@mfs.iitr.ac.in*
*Indian Institute of Technology, Roorkee*

**Agam Pandey**[*]                                                        *agam_p@ce.iitr.ac.in*
*Indian Institute of Technology, Roorkee*

**Amritanshu Tiwari**[*]                                              *amritanshu_t@mfs.iitr.ac.in*
*Indian Institute of Technology, Roorkee*

**Sukrit Jindal**[*]                                                      *sukrit_j@mfs.iitr.ac.in*
*Indian Institute of Technology, Roorkee*

**Swadesh Swain**[*]                                                    *swadesh_s@ece.iitr.ac.in*
*Indian Institute of Technology, Roorkee*

**Reviewed on OpenReview:** *https://openreview.net/forum?id=5L9Ocl0xtf*

## Abstract

Large Vision-Language Models (VLMs) have revolutionized computer vision, enabling tasks such as image classification, captioning, and visual question answering. However, they remain highly vulnerable to adversarial attacks, particularly in scenarios where both visual and textual modalities can be manipulated. In this study, we conduct a comprehensive reproducibility study of *"An Image is Worth 1000 Lies: Adversarial Transferability Across Prompts on Vision-Language Models"* validating the Cross-Prompt Attack (CroPA) and confirming its superior cross-prompt transferability compared to existing baselines. Beyond replication we propose several key improvements: (1) A novel initialization strategy that significantly improves Attack Success Rate (ASR). (2) Investigate cross-image transferability by learning universal perturbations. (3) A novel loss function targeting vision encoder attention mechanisms to improve generalization. Our evaluation across prominent VLMs—including Flamingo, BLIP-2, and InstructBLIP as well as extended experiments on LLaVA validates the original results and demonstrates that our improvements consistently boost adversarial effectiveness. Our work reinforces the importance of studying adversarial vulnerabilities in VLMs and provides a more robust framework for generating transferable adversarial examples, with significant implications for understanding the security of VLMs in real-world applications. Our code is available at https://github.com/Swadesh06/Revisting_CroPA

## 1 Introduction

The advent of large Vision-Language Models (VLMs) has significantly transformed the field of computer vision by enabling a wide range of tasks, including image classification, captioning, and visual question answering. This versatility has fostered deeper exploration into visual-linguistic interactions. However, recent studies Zhao et al. (2023); Qi et al. (2023); Zhang et al. (2022a); Carlini et al. (2024) have demonstrated that VLMs remain highly vulnerable to adversarial attacks. These attacks involve subtle perturbations to

---

[*]All authors contributed equally

input images, leading VLMs to produce incorrect or even harmful outputs. Furthermore, the inclusion of textual modalities introduces additional attack vectors, expanding the range of threats beyond those faced by traditional vision models.

Several studies have investigated the adversarial robustness of VLMs. For example, Zhao et al. (2023) conducted a comprehensive analysis of the adversarial robustness of VLMs such as BLIP and BLIP-2, exploring both query-based and transfer-based adversarial attack methods in black-box settings. Additionally, Schlarmann & Hein (2023) examined targeted and untargeted adversarial attacks in white-box settings. While these works primarily focused on adversarial image attacks, subsequent research has also explored adversarial perturbations in textual inputs. Qi et al. (2023) demonstrated that adversarial images could manipulate VLMs into executing harmful instructions, while Tu et al. (2023) systematically evaluated both visual and textual adversarial attacks.

Traditionally, the generalization of adversarial examples and their transferability in VLMs has been classified into two primary categories:

- **Cross-Model Transferability**: The ability of adversarial examples to maintain their adversarial nature across different VLM architectures, commonly referred to as transferability.

- **Cross-Image Transferability**: The ability of adversarial perturbations to generate adversarial examples that generalize across multiple images, often known as Universal Adversarial Perturbations (UAPs).

Luo et al. (2024) introduced the novel concept of **Cross-Prompt Transferability**, which describes the ability of adversarial images to remain effective across varying textual prompts. Unlike prior work that treated visual and textual adversarial perturbations independently, Luo et al. (2024), proposed the **Cross-Prompt Attack** (CroPA), which employs learnable prompts to ensure adversarial images retain their effectiveness regardless of textual input. Their work demonstrated CroPA's efficacy across multiple vision-language tasks, including image classification, captioning, and visual question answering.

Our work aims to address the following goals:

- [**Reproducibility Study**] **Reproducing the Results from the Original Paper:** Through our experiments, we successfully reproduced and verified the four main claims of the original paper. First, we confirm that CroPA achieves superior cross-prompt transferability compared to established baselines across various target texts, and that this transferability is independent of the semantic meaning or word frequency of the target text. Second, we show that while increasing the number of prompts improves transferability, convergence occurs rapidly, and even with additional prompts, baseline methods fail to surpass CroPA. Third, we validate CroPA's effectiveness even when additional in-context learning examples are provided. Lastly, we reproduce the original findings on CroPA's convergence by evaluating the overall ASR across increasing attack iterations, while also contributing our own analysis of ASR performance for individual tasks.

- [**Extended Work**] **Better Initialization Strategy:** We propose a new initialization method, substantially increasing the Attack Success Rate (ASR) as well as Transferability.

- [**Extended Work**] **Investigating Cross Image Transferability and Image Augmentation:** We explore learning a common perturbation for all images using CroPA's backbone and investigate whether transferability-promoting enhancements aid in the cause.

- [**Extended Work**] **Guiding perturbations via Target Value Vectors:** We propose a novel loss function building on the idea that specific components within the vision encoder's attention mechanism control and determine the level of interaction between patches, manipulating the value vectors of the vision encoder in a targeted manner leads to greater generalization as well as ASR.

Furthermore, we conduct in-depth analyses to elucidate the mechanisms behind our improvements, offering insights into the nature of adversarial vulnerabilities in VLMs. Our work not only reinforces the importance

of studying these vulnerabilities but also provides a more robust and versatile framework for generating transferable adversarial examples.

## 2 Related Works

While adversarial attacks on deep neural networks have been extensively studied since their introduction by Szegedy et al. (2013) and Goodfellow et al. (2014), their application to Vision-Language Models (VLMs) presents unique challenges and opportunities. In this section, we review the progression of adversarial transferability research and position CroPA within this broader context.

### 2.1 Adversarial Transferability

The phenomenon of adversarial transferability has manifested across multiple dimensions in machine learning systems. Liu et al. (2017) and Tramèr et al. (2017) established that adversarial examples crafted for one model can often deceive other models, even those with different architectures. Beyond model transferability, researchers discovered that perturbations can generalize across images Moosavi-Dezfooli et al. (2017); Mopuri et al. (2017), enabling the creation of Universal Adversarial Perturbations (UAPs).

The domain of transferability extends to cross-task applications, where adversarial examples designed for classification can affect object detection and other visual tasks Naseer et al. (2019b;a); Lu et al. (2020); Salzmann et al. (2021). This multi-dimensional transferability raises critical questions about the robustness of modern machine learning systems.

### 2.2 Adversarial Attacks on Vision-Language Models

Early research on adversarial robustness of vision-language models focused on task-specific attacks. In image captioning, works by Xu et al. (2019); Zhang et al. (2020); Aafaq et al. (2021); Chen et al. (2017) demonstrated methods to manipulate model outputs. Similarly, Visual Question Answering (VQA) systems proved vulnerable to targeted attacks that mislead attention mechanisms Xu et al. (2018); Kaushik et al. (2021); Kovatchev et al. (2022); Li et al. (2021); Sheng et al. (2021); Zhang et al. (2022b).

However, these approaches primarily targeted lightweight CNN-RNN architectures lacking in-context learning capabilities, limiting their applicability to contemporary VLMs. Recent work by Zhao et al. (2023) explored adversarial robustness of modern large VLMs like BLIP and BLIP-2 under black-box settings, focusing on cross-model transferability rather than the cross-prompt perspective introduced by CroPA.

### 2.3 Cross-Prompt Adversarial Transferability

The emergence of large VLMs with prompt-driven task adaptation introduced a new dimension of adversarial vulnerability. Unlike traditional vision models that perform specific tasks, VLMs can dynamically adapt to different tasks based on textual prompts Li et al. (2023b); Awadalla et al. (2023); Li et al. (2023b); Zhu et al. (2023); Liu et al. (2023). This flexibility creates a novel attack surface where adversarial examples must maintain their effectiveness across varying prompts.

CroPA Luo et al. (2024) pioneered this concept by introducing learnable prompts that compete with image perturbations during optimization. This approach diverges from existing methods that treat visual and textual perturbations independently, establishing a new paradigm for evaluating VLM robustness. Due to the relative novelty of this concept, no credible benchmarks/baselines have been established yet. However, work by Yang et al. (2024) does explore cross-prompt transferability and introduces Context Injection Attack (CIA), which employs gradient-based perturbations to inject target tokens into both visual and textual contexts, and is presented as an additional comparison in our experiments.

### 2.4 Enhancing Adversarial Transferability

Various techniques have been proposed to improve adversarial transferability. Input diversity methods Xie et al. (2019) and data augmentation strategies Wang et al. (2021) have shown promise in creating more

robust attacks. Recent works have identified gradient misalignment as a key factor limiting transferability Li et al. (2023a); Wang et al. (2023), leading to novel optimization strategies. Our work builds upon these insights to enhance CroPA's capabilities across multiple transferability dimensions.

## 3 Scope of reproducibility

This study aims to examine and validate the results demonstrated by Luo et al. (2024). Our primary objective is to confirm that CroPA significantly enhances the transferability of adversarial examples across various prompts by meticulously reproducing their experimental procedures.

The main claims we aim to verify are as follows:

- **Claim 1:** CroPA achieves cross-prompt transferability across various target texts

- **Claim 2:** CroPA achieves the best overall performance across different number of prompts

- **Claim 3:** CroPA converges towards higher ASR as number of iterations are increased

- **Claim 4:** CroPA outperforms baselines under few-shot settings

Beyond replication, we extend the scope of CroPA by investigating its efficacy in cross-model and cross-image contexts as well as compare it to other methods in cross-prompt transferability. Specifically, we assess whether adversarial images generated through CroPA can consistently deceive diverse Vision-Language Models (VLMs), regardless of the input prompt or specific model parameters. Addressing these points will enable us to faithfully reproduce the original paper's experiments and build upon it to explore the broader applicability of CroPA in enhancing the robustness and versatility of adversarial attacks on VLMs.

## 4 Methodology

### 4.1 Problem Formulation

The vulnerability of neural networks to adversarial attacks has been well-documented since the seminal work of Goodfellow et al. (2014). Building on this foundation, we examine the authors' novel formulation that extends these concepts to cross-prompt scenarios in Vision-Language Models (VLMs). Their work introduces a critical perspective on how adversarial perturbations can maintain effectiveness across varying textual inputs Luo et al. (2024).

The authors develop their formulation around a VLM function $f$ that processes both visual and textual inputs, denoted as $x_v$ and $x_t$ respectively. To ensure real-world applicability, the authors constrain the adversarial perturbation $\delta_v$ within human-imperceptible bounds, enforcing $\|\delta_v\|_p \leq \epsilon_v$. This constraint mirrors established practices in adversarial machine learning while adapting them to the multi-modal context of VLMs Carlini et al. (2024).

The authors establish two distinct attack scenarios that we reproduced in our study. (1) The **targeted attack** scenario aims to manipulate the VLM into generating a specific predetermined text $T$, regardless of the input prompt. This objective manifests mathematically as Equation 1 minimizing the language modeling loss $L$ across multiple prompt instances. In contrast, (2) the **non-targeted** scenario focuses on maximizing the discrepancy between outputs from clean and adversarial inputs, described in Equation 2.

$$\min_{\delta_v} \sum_{i=1}^{k} L(f(x_v + \delta_v, x_t^i), T) \tag{1}$$

$$\max_{\delta_v} \sum_{i=1}^{k} L(f(x_v + \delta_v, x_t^i), f(x_v, x_t^i)) \tag{2}$$

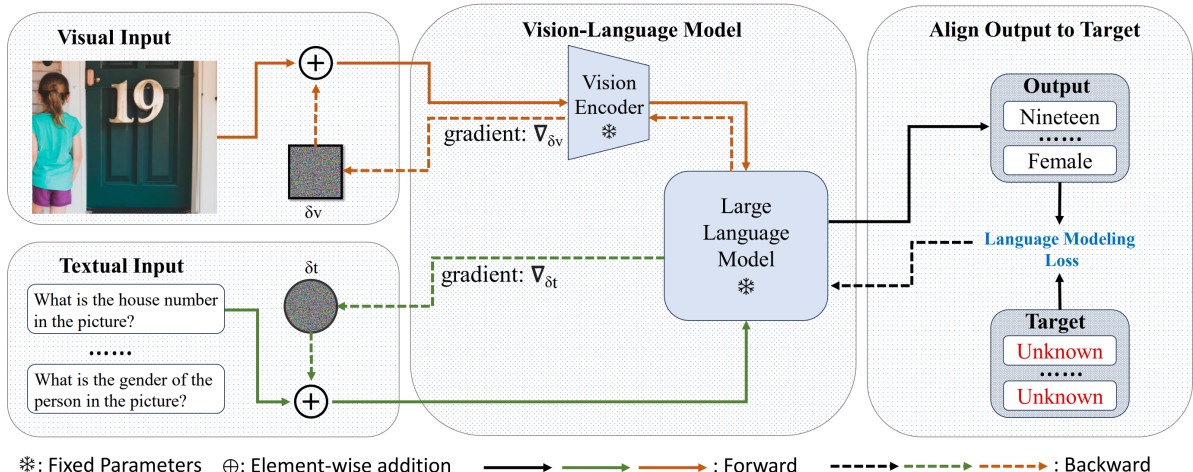

Figure 1: Framework overview of CroPA as presented in Luo et al. (2024). The architecture employs learnable perturbations for both image ($\delta_v$) and prompt ($\delta_t$) inputs. These perturbations operate antagonistically, while $\delta_v$ is optimized to minimize the language modeling loss, $\delta_t$ works to maximize it. The model's forward pass (solid arrows) processes the perturbed inputs through the vision encoder and language model, with backpropagation (dashed arrows) updating the perturbations at configurable frequencies. The model parameters (marked with *) remain fixed during the attack. $\oplus$ denotes element-wise addition.

The effectiveness of these attacks is quantified through the Attack Success Rate (ASR). For targeted attacks, success requires generating the exact target text, while non-targeted attacks succeed by producing any output that differs from the clean image's prediction.

## 4.2 Baseline Approaches

Given that cross-prompt transferability represents a novel direction in adversarial machine learning, no established baselines exist in current literature. Therefore, we examine two baseline approaches introduced by Luo et al. (2024) to evaluate CroPA's effectiveness: Single-P and Multi-P.

Single-P represents the simplest baseline, where an image perturbation is optimized using a single prompt. The optimization objective for targeted attacks is:

$$\min_{\delta_v} L(f(x_v + \delta_v, x_t), T) \tag{3}$$

Multi-P extends this concept by utilizing multiple prompts during optimization. Given a collection of textual prompts $X_t = \{x_t^1, x_t^2, ..., x_t^k\}$, the objective becomes:

$$\min_{\delta_v} \sum_{i=1}^{k} L(f(x_v + \delta_v, x_t^i), T) \tag{4}$$

For non-targeted attacks, these objectives are inverted to maximize the discrepancy between the model's outputs for clean and perturbed inputs. The effectiveness of both approaches serves as a reference point for evaluating CroPA's enhanced cross-prompt transferability.

### 4.3 Cross Prompt Attack

The Cross-Prompt Attack (CroPA) method introduced by Luo et al. (2024) employs learnable prompts during optimization to enhance cross-prompt transferability. The key innovation lies in using prompt perturbations that compete with image perturbations during the optimization process rather than collaborating to deceive the model.

The algorithm optimizes both visual perturbation $\delta_v$ and prompt perturbation $\delta_t$ but with opposing objectives. While $\delta_v$ aims to minimize the language modeling loss for generating the target text, $\delta_t$ maximizes this loss. This adversarial relationship between the perturbations forces $\delta_v$ to develop stronger transferability across different prompts.

The optimization process can be formally expressed as a min-max problem:

$$\min_{\delta_v} \max_{\delta_t} L(f(x_v + \delta_v, x_t + \delta_t), T) \tag{5}$$

where $f$ represents the VLM, $T$ is the target text for targeted attacks, and $L$ denotes the language modeling loss.

The implementation follows an iterative approach using Projected Gradient Descent (PGD) Madry et al. (2017). The visual perturbation updates use gradient descent to minimize the loss, while prompt perturbation updates employ gradient ascent to maximize it. The update frequency can be controlled via a parameter $N$, where image perturbation updates occur $N$ times for each prompt perturbation update. The framework can be visualized formally in Figure 1. We reproduce the algorithm, described by Luo et al. (2024), as follows:

---

**Algorithm 1** CroPA: Cross Prompt Attack

---

**Require:** Model $f$, Target Text $T$, vision input $x_v$, prompt set $X_t$, perturbation size $\epsilon$, step sizes $\alpha_1$ and $\alpha_2$, number of iteration steps $K$, adversarial prompt update interval $N$

**Ensure:** Adversarial example $x_v'$

1: Initialize $x_v' = x_v$
2: **for** step $= 1$ to $K$ **do**
3:     Uniformally sample the prompt $x_t^i$ from $X_t$
4:     **if** $x_t^i$ is not initialised **then**
5:         Initialise $x_t'^i = x_t^i$
6:     **end if**
7:     Compute gradient for adversarial image : $g_v = \nabla_{x_v} L(f(x_v', x_t^i), T)$
8:     Update with gradient descent: $x_v' = x_v' - \alpha_1 \cdot \text{sign}(g_v)$
9:     **if** $\text{mod}(step, N) = 0$ **then**
10:        Compute gradient for adversarial prompt: $g_t = \nabla_{x_t} L(f(x_v', x_t^i), T)$
11:        Update with gradient ascent: $x_t'^i = x_t'^i + \alpha_2 \cdot \text{sign}(g_t)$
12:     **end if**
13:     Project $x_v'$ to be within the $\epsilon$-ball of $x_v$: $x_v' = \text{Clip}_{x_v, \epsilon}(x_v')$
14: **end for**
15: **return**   $x_v'$

---

During evaluation, only the optimized image perturbation is applied, while prompt perturbations are discarded. This ensures that the attack's effectiveness stems from the image perturbation's inherent transferability rather than prompt modifications.

## 5 Additional Investigative Experiments

CroPA's iterative optimization method using PGD under the hood to increase the attack success rate is effective when used on single model, but the use of the dependency on the feedback of single model minimizing the loss function limits it's transferability, as is shown true for several other gradient based perturbation

techniques, which many works have recognized and proposed enhancements to Qin et al. (2024); Li et al. (2024); Wang et al. (2023); Li et al. (2023a). This limitation is also recognized by the authors in their original paper, and it verified by Table 9 under Appendix A.1. Furthermore, as reported in Table 1, the performance of CroPA is especially poor in certain tasks such as Classification and Captioning.

Additionally, CroPA is highly computationally intensive, taking over 6 hours to run on a 48GB L40S GPU utlising 95% of the VRAM, wherein a mere 50 images are trained with adversarial perturbations.

These limitations open the scope to perform our extended experiments which build on the CroPA framework and supplement these drawbacks incorporating novel enhancements, especially targeting : (i) enhancements in CroPA's lack of consistent performance across all tasks and (ii) investigating CroPA against the different transferability concepts we mentioned in Section 1, and incorporate supplements into the original framework move towards better transferability of image perturbations.

While not a core proposal of the Luo et al. (2024) exploring transferability will also aid in cutting down computation costs, with cross-image transferable perturbations having the potential to significantly increase efficiency in an ideal scenario, without having to train a separate image perturbation for each image. The Cross-Model transferable perturbation can potentially achieve even greater efficiency.

The following subsections detail our experiments which underscore our motivations to achieving the aforementioned goals and outline their individual mechanisms:

### 5.1 Noise Initialization via Vision Encoding Optimization

A core limitation of CroPA lies in its random noise initialization strategy, which is shared across many gradient-based adversarial attack methods. While CroPA iteratively optimizes this noise, its starting point is uninformed by the target semantics, leading to instability and slower convergence. This is especially problematic when targeting vision-language models (VLMs), where semantic alignment with prompts is crucial. Crucially, this limitation is not merely theoretical, our comprehensive evaluation of CroPA's bi-level optimization framework in Section 6.5.1 reveals that its effectiveness is highly sensitive to initialization, as empirically verified in Figure 7 and Table 6.5.1. This sensitivity stems from CroPA's inability to guide the optimization trajectory toward semantically meaningful directions from the outset, highlighting the need for a more principled initialization strategy. Furthermore, conventional initialization techniques fail to ensure meaningful adversarial directions from the onset, leading to suboptimal attack success rates and reduced cross-prompt transferability.

To address these limitations, we hypothesize that a semantically informed initialization of adversarial perturbations rather than a purely random initialization can significantly enhance attack efficacy while maintaining perturbation imperceptibility. Specifically, we propose leveraging a diffusion-based approach to synthesize a target image corresponding to a desired prompt. By using the vision encoder's feature space as an alignment mechanism, we ensure that the initial perturbation is already oriented in a semantically meaningful direction.

Our proposed Noise Initialization via Vision Encoding Optimization method integrates this semantic anchoring with adversarial optimization by:

- Generating a semantically relevant target image using a state-of-the-art diffusion model (e.g., Stable Diffusion XL).

- Aligning the perturbation with the target image's vision encoder representation, ensuring that the initial noise conforms to the adversarial objective.

- Optimizing perturbation updates using a Mean Squared Error(MSE)-based loss function that enforces semantic similarity while maintaining adversarial effectiveness.

By replacing random initialization with this vision-guided approach, we not only improve adversarial robustness but also enhance attack efficiency, reducing the number of optimization steps required for convergence. Our method thus provides a principled alternative to conventional approaches, offering both stronger ad-

versarial perturbations and improved cross-prompt transferability. This approach provides a more effective initialization, ensuring that the adversarial perturbations are semantically informed from the start.

### 5.1.1 Noise Initialization

Preliminary experiments performed on models such as BLIP-2 and InstructBLIP suggest that initializing adversarial perturbations with semantically informed noise significantly enhances the attack's efficacy over traditional random initializations while only increasing the computation time for a single image by 20-25 seconds on a single GPU. The diffusion-based approach not only improves alignment in the vision encoder's feature space but also preserves the visual fidelity of the resulting adversarial examples while conforming to strict perturbation budgets.

### 5.1.2 Diffusion-Based Target Synthesis

Given a target prompt $T$, we first generate an image $\mathbf{x}_{\text{target}}$ that embodies the semantic attributes described by $T$. This is accomplished via a diffusion model, specifically Stable Diffusion XL (SDXL). The generation process can be formalized as follows:

$$\mathbf{x}_{\text{target}} = \mathcal{D}(T, \mathbf{z}; \theta_{\text{SDXL}}), \tag{6}$$

where $\mathcal{D}$ denotes the diffusion process, $\mathbf{z}$ is sampled from a Gaussian distribution, and $\theta_{\text{SDXL}}$ represents the pre-trained weights of the diffusion model. The resulting image $\mathbf{x}_{\text{target}}$ effectively captures the semantic essence of the prompt $T$, thus providing an informative basis for initializing adversarial perturbations.

### 5.1.3 Vision-Encoder Anchored Perturbation

Let $f_v(\cdot)$ denote the vision encoder component of the VLM. Our objective is to craft an adversarial perturbation $\boldsymbol{\delta}$ such that the perturbed image $\mathbf{x} + \boldsymbol{\delta}$ mimics the semantic representation of the target image in the vision encoder's output space. To achieve this, we derive the initial perturbation by minimizing the following objective:

$$\boldsymbol{\delta}_{\text{init}} = \arg\min_{\|\boldsymbol{\delta}\|_\infty \leq \epsilon} \|f_v(\mathbf{x} + \boldsymbol{\delta}) - f_v(\mathbf{x}_{\text{target}})\|_2^2, \tag{7}$$

where $\epsilon$ is the maximum allowable perturbation (ensuring imperceptibility under an $\ell_\infty$ constraint). This initialization ensures that the adversarial example starts within a semantically meaningful neighborhood of the target prompt's representation. The method is formally depicted in Algorithm 2 under Appendix B.

### 5.1.4 Adversarial Optimization via PGD

After initializing the perturbation, we refine the adversarial example using projected gradient descent (PGD). For the $t^{\text{th}}$ iteration, the update rule is given by:

$$\mathbf{x}_{\text{adv}}^{(t+1)} = \Pi_{\mathcal{B}_\epsilon(\mathbf{x})}\left[\mathbf{x}_{\text{adv}}^{(t)} - \alpha \nabla_{\mathbf{x}_{\text{adv}}^{(t)}} \mathcal{L}_{\text{MSE}}\right], \tag{8}$$

where $\alpha$ is the step size, and $\Pi_{\mathcal{B}_\epsilon(\mathbf{x})}$ projects the updated input back onto the admissible $\ell_\infty$ ball around the original image $\mathbf{x}$. The loss function used during optimization is a simple mean squared error (MSE) between the vision encoder outputs:

$$\mathcal{L}_{\text{MSE}} = \|f_v(\mathbf{x}_{\text{adv}}) - f_v(\mathbf{x}_{\text{target}})\|_2^2. \tag{9}$$

This loss ensures that each update incrementally aligns the adversarial example with the target's semantic embedding.

## 5.2 Guiding perturbations via Target Value Vectors and evaluating Cross-Model Tranferability

While CroPA attempts to enhance adversarial effectiveness by optimizing both image and text inputs through PGD, it lacks architectural sensitivity, particularly to the internal dynamics of the vision encoder, which plays a dominant role in VLMs' semantic grounding. CroPA treats both vision and language modalities as uniformly vulnerable, failing to exploit modality-specific weaknesses. This oversight limits its ability to

generate transferable and structured adversarial perturbations. More critically, CroPA's perturbations lack targeted influence on the encoder's representational backbone, which undermines both adversarial strength and transferability across models. CroPA minimizes and maximizes losses for image and text perturbations respectively, aiming to manipulate the model's output toward a specific target.

Previous research (Cui et al. (2023); Wang et al. (2024)) demonstrates that perturbing the output embeddings of the vision encoder can significantly impair the visual perception capabilities of large vision-language models (LVLMs), thereby misleading the language model's responses. Moreover, various LVLMs typically employ similar image encoders to convert images into visual tokens, which encode rich semantic and structural information and serve as a critical interface for language decoders, to access and reason over image content. However, the CroPA method does not sufficiently leverage the architectural specifics of the vision encoder and instead treats both the vision and language components as equally susceptible to adversarial manipulation, potentially overlooking targeted vulnerabilities unique to the vision encoder.

To address this, we propose integrating the CroPA loss with a perturbation loss designed to disrupt the vision encoder's core attention mechanism, specifically focusing on value vectors of the attention layers of the vision encoder. *Our hypothesis is that by perturbing the value vectors between the original image and a reference image corresponding to a target text, we can significantly enhance the Adversarial Success Rate (ASR) while maintaining the same hyperparameter constraints as CroPA.*

This motivation is derived from empirical evidence in related works on adversarial tranferability in VLMs. Our approach is inspired by the Doubly-Universal Adversarial Perturbation (Doubly-UAP) method where instead of maximizing, we minimize the losses between value vectors of $x_i$ and $T_i$, which identifies the most effective components within a vision encoder's attention mechanism for adversarial influence. We hypothesize that targeting the value vectors in the vision layers which encode essential visual features will disrupt the model's interpretative abilities more effectively than CroPA's generic PGD-based perturbations. Additionally, by freezing the text encoder, we ensure that our perturbations are specifically tuned to degrade the visual processing capability of the VLM rather than introducing text-based artifacts.

Thus, our modification aims to:

1. Improve attack effectiveness by focusing on exploiting information in the vision encoder,

2. enhance semantic alignment of adversarial perturbations with the target text representation, and

3. achieve better adversarial robustness across models while maintaining computational feasibility.

By jointly optimizing the CroPA loss and our modified Doubly-UAP loss, we introduce a more structured, interpretable, and targeted adversarial attack framework for VLMs.

This incorporation is adapted from a component of the Doubly-UAP method proposed in Kim et al. (2024) which introduced a UAP by identifying which specific components within the vision encoder's attention mechanism most effectively influence the performance of the VLM. We choose to focus on the two components with the most fundamental roles in the attention mechanism:

1. **Attention Weights**: These control how much each patch should focus on other patches, determining the level of interaction or relevance between patches. We hypothesize that by targeting the attention weights, we can effectively interfere with the encoder's ability to establish these relationships.

2. **Value Vectors**: These hold the actual information within each patch. We expect that perturbing the value vectors will disrupt the essential information content within patches, further impairing the model's interpretative abilities.

We target the vision encoders within VLM, as they are crucial for visual interpretation. Specifically, we focus on the attention mechanism within the vision encoder, the core process responsible for interpreting visual features. We aim to target the most vulnerable components of this mechanism the value vectors at the middle-to-late layers based on prior analysis.

Formally, in the standard Doubly-UAP attack, the perturbation $\delta^*$ is obtained as:

$$\delta^* = \arg\max_\delta \frac{1}{|L|} \sum_{l \in L} \text{Loss}(V_l(x), V_l(x + \delta)), \tag{10}$$

where $V_l(x)$ represents the value vectors associated with the $l$-th layer with input image $x$, and $\text{Loss}(\cdot)$ is the loss function applied to the target vectors.

### 5.2.1 Our Modified Approach

We adapt this approach by introducing a modified loss function that incorporates a target value vector derived from a reference image corresponding to a desired target text **T**. Instead of solely maximizing the deviation of value vectors from their original representation, we enforce alignment between the vision encoder's output and a predefined target representation. Our method consists of the following steps:

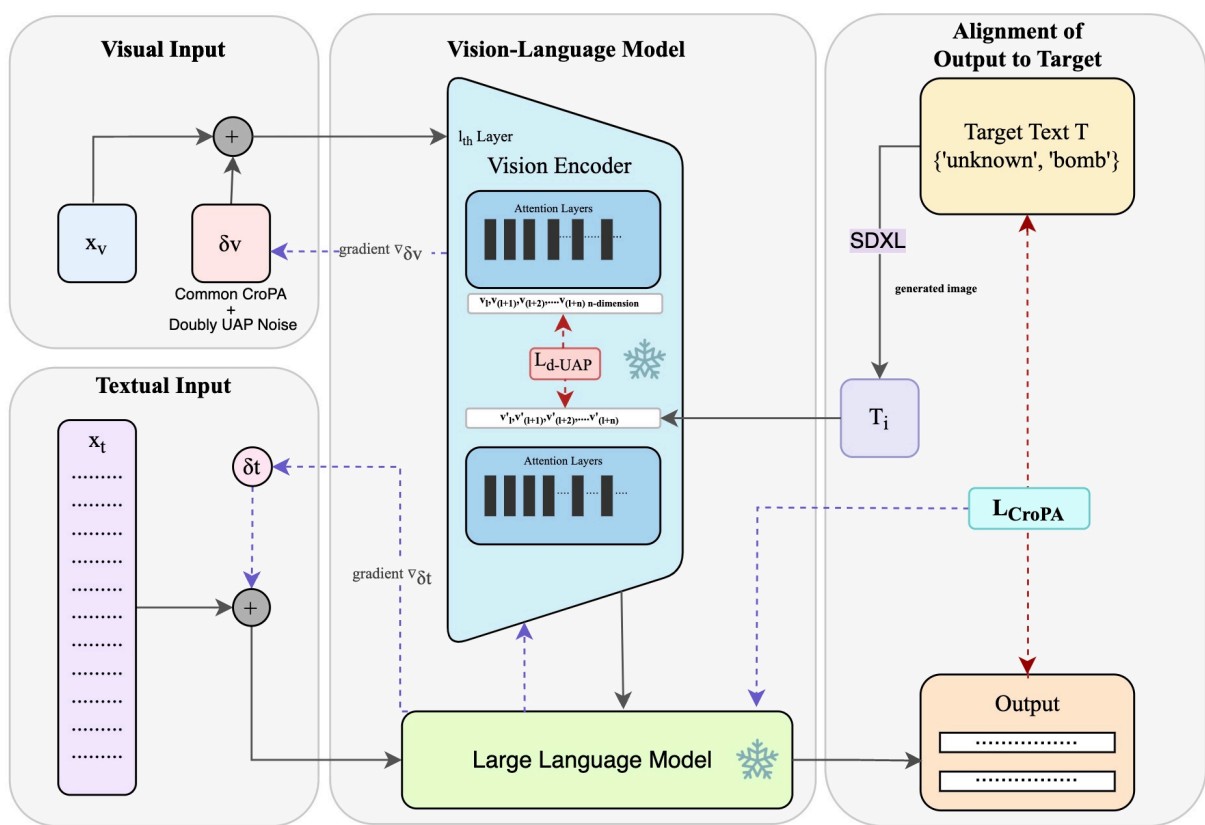

Figure 2: D-UAP Architecture Diagram

1. Extract the value vectors from the $l$-th layer of the vision encoder for both the input image and the target text's associated image.

2. Compute the cosine similarity loss between these value vectors.

3. Jointly minimize -ve of this loss along with the CroPA loss with proper scaling to ensure adversarial robustness and target alignment.

Formally, let $V_i(x + \delta)$ denote the perturbed value vectors of the $i$-th attention head for the input image $x$, and let $V_t$ represent the value vectors of the target text's reference image. We define our value vector loss as:

$$L_{\text{d-UAP}} = \sum_{i=1}^{N} \left( 1 - \frac{V_i(x + \delta) \cdot V_t}{\|V_i(x + \delta)\|_2 \|V_t\|_2} \right) \tag{11}$$

where $N$ is the number of attention heads. This loss encourages the perturbed image's value vectors to closely align with the target text's representation.

We integrate this loss with the CroPA loss to formulate our final objective function in Equation 12 , where $\lambda$ is a hyperparameter controlling the relative importance of the value vector loss:

$$L_{\text{re-CroPA}} = L_{\text{CroPA}}(x_v + \delta_v, x_t + \delta_t, T) - \lambda L_{\text{d-UAP}}(\delta_v) \tag{12}$$

By jointly optimizing both losses, our approach not only preserves the adversarial nature of the perturbation but also enforces semantic alignment with the target text. Specifically, during optimization, the gradients from both loss components are combined to update the perturbation $\delta$, ensuring that the generated adversarial example exhibits both cross-prompt transferability and guided semantic influence. This enhancement to the Doubly-UAP framework allows for more precise adversarial manipulation of VLMs, facilitating controlled and interpretable perturbations with applications in adversarial robustness and security analysis. Algorithm 3 under Appendix B summarizes the method step-by-step.

While we propose this as a step towards increasing transferability of perturbations generated by CroPA, the dependency of the D-UAP method on the model value vectors, and consequently the Vision Encoder, leads us to hypothesize that transferability might be limited to models having the same or similar Vision Encoder architectures as the primary model on which the CroPA framework is run.

### 5.3 Transferability across images via Image Augmentation

Traditional adversarial methods optimize perturbations through iterative gradient descent on individual images, often leading to overfitting to image-specific features. As a result, these perturbations exhibit poor transferability across images, which makes them computationally expensive. The same applies to CroPA, in which the adversarial perturbations are learned for a single image, across multiple prompts. In their experiments, Luo et al. (2024) mention that their method is orthogonal to cross-image transferability approaches, and that that cross-image transferability can be enhanced by computing perturbations across images, as is done in Moosavi-Dezfooli et al. (2017). As part of our investigation, we experiment on this claim and also introduce a novel approach, based on other literature on Universal Adversarial Perturbations (UAPs).

To mitigate overfitting in gradient-based attacks while enabling cross-image transferability, Fang et al. (2024) introduce Universal Adversarial Perturbations (UAPs) for Vision-Language Pretraining (VLP) models. Complementing this, CutMix, proposed by Yun et al. (2019), offers a distinct regularization strategy by synthesizing training samples through regional replacement rather than pixel blending. CutMix generates composite images by cutting rectangular patches from one image and pasting them into another, while mixing labels proportionally to the area of the patches. This forces models to learn from partial views and prioritize spatially invariant features, thereby improving robustness against adversarial attacks and occlusions.

Expanding on UAP frameworks, Zhang et al. (2024) propose Effective and Transferable Universal Adversarial Attack (ETU), a method that enhances UAP transferability across multiple VLP models and tasks through improved global and local optimization techniques. Specifically, ETU introduces SCMix, a data augmentation strategy combining self-mix and cross-mix operations to boost data diversity while preserving semantic integrity. Of these, SCMix aims to integrate both spatial and scale invariant features in images. We integrate SCMix into our perturbation generation process, selecting it as one of our investigative techniques based on their empirical results showing that, for the same model, SCMix alone, without their additional local utility reinforcement, yields superior cross-image transferability.

### 5.3.1 SCMix

SCMix, as an augmentation technique, diversifies training inputs through a structured two-stage mixing process, the first of which self-mixing and the second cross-mixing. The self-mixing phase synthesizes variations of a base image $I_1$ by blending randomly cropped and resized patches:

$$I'_1 = \eta X_1 + (1 - \eta)X_2 \quad \text{where} \quad X_i = \text{Resize}(\text{RandomCrop}(I_1)), \quad \eta = 0.5 \tag{13}$$

While the original SCMix method utilised a probabilistic sampling of $\eta$, as $\eta \sim \text{Beta}(\alpha, \alpha)$ for some $\alpha > 0$, we choose $\eta = 0.5$ to be fixed, which is equivalent to setting $\alpha = \infty$ which is justified in appendix D.3. The subsequent cross-mixing phase introduces features from a secondary image $I_2$ through a weighted combination:

$$I_3 = \beta_1 I'_1 + \beta_2 I_2 \quad \text{with} \quad \beta_1 \gg \beta_2 \in [0, 1). \tag{14}$$

The weighting ensures $I_1$'s dominant visual semantics persist in $I_3$ while incorporating diversity from elements of $I_2$, enabling the adversarial generator to learn perturbations effective across diverse images. Further, by preserving the dominance of the base image, the compatibility with $I_1$'s text description $T_1$ for $I_3$ is ensured. This facilitates enriched cross-modal interactions, as the synthetic image $I_3$ remains aligned with $T_1$ despite integrating $I_2$'s features.

The dual mixing mechanism reduces overfitting which is possible in the case of CroPA by forcing perturbations to attack features invariant to both intra-image variations (via self-mixing) and inter-image blending (via cross-mixing). This regularization effect promotes universal perturbations for cross-image transferability, which may be applicable to unseen images without explicitly training the attack for that image, bridging the gap between targeted efficacy and cross-image generalization. This is valid because during inference, CroPA only returns the perturbed image, or equivalently an appropriate perturbation for the image, thus, if effective cross-image perturbations are learned, the attack can be made universal as being both cross-prompt transferable, as CroPA claims, and cross-image transferable, as we aim to check.

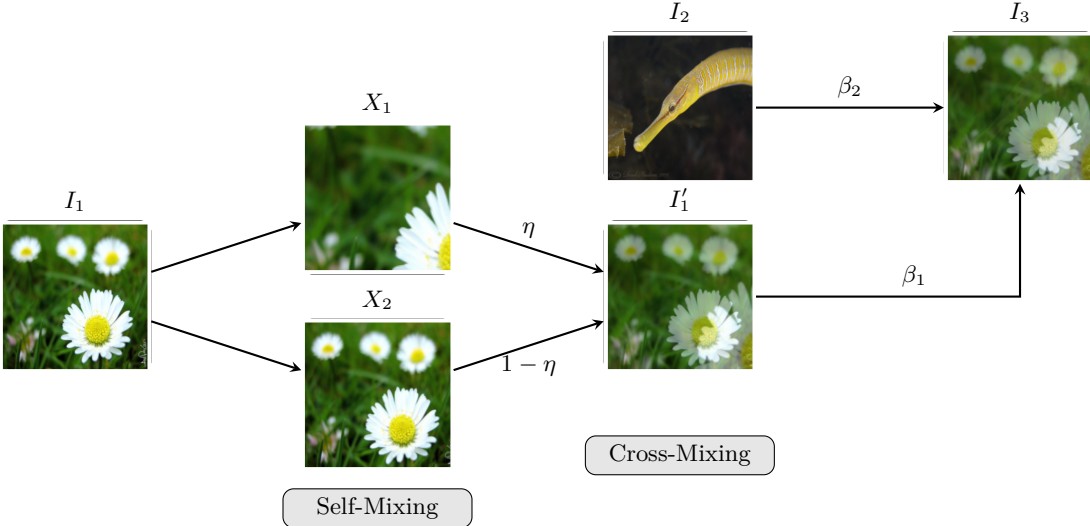

Figure 3: Illustration of the self-mixing and cross-mixing processes. The input image $I_1$ undergoes self-mixing to generate intermediate representations $X_1$ and $X_2$, which are then combined to form $I'_1$. Additionally, an external image $I_2$ is mixed with $I'_1$ to produce the final output $I_3$.

### 5.3.2 CutMix

CutMix Yun et al. (2019) enhances transferability through discrete regional feature replacement. For image pairs $x_A$ and $x_B$, it creates composite samples:

$$\tilde{x} = M \odot x_A + (1 - M) \odot x_B \tag{15}$$

where $M \in \{0, 1\}^{W \times H}$ is a binary mask defining a randomly placed rectangular region. Effectively, CutMix is able to learn from partial views of the images that it mixes. The main difference with SCMix is that while SCMix uses a soft blend of features across, and within, images, CutMix uses a hard mask-based replacement. Unlike SCMix's continuous blending, CutMix's discrete region replacement forces perturbations to target spatially invariant features that persist across different image combinations. When integrated with adversarial training, this approach encourages perturbations that exploit higher-level semantic patterns rather than localized artifacts, potentially improving cross-image transferability.

### 5.4 Attack Setup and Threat Models for Transferability Experiments

We establish comprehensive threat models for evaluating both cross-model and cross-image transferability, focusing on the specific adversarial capabilities, knowledge assumptions, and evaluation protocols for each scenario.

### 5.4.1 Using D-UAP to explore Cross-Model Transferability for CroPA

To systematically evaluate how well adversarial perturbations generated with our D-UAP enhancement transfer across different models, we establish a principled threat model for cross-model attacks:

**Adversary's Goal:** The attacker aims to generate a single adversarial perturbation for an image that, when applied, causes multiple vision-language models to output a specific target text regardless of the input prompt. This represents a practical black-box transfer scenario where an attacker optimizes on an accessible model but deploys against unknown models.

**Knowledge and Capabilities:** The attacker has white-box access to a surrogate model (either BLIP-2 or Flamingo in our experiments), including gradients and internal representations. For target models, the attacker only knows their general architecture family (e.g., CLIP-based) without access to parameters. The attacker is constrained to a maximum perturbation budget of $\epsilon = 16/255$ under $L_\infty$ norm to ensure visual imperceptibility.

**Cross-Model Settings:** We examine two transferability scenarios reflecting real-world deployment conditions:

- **Intra-family transfer:** Perturbations optimized on BLIP-2 (with OPT-2.7b language model) are evaluated on InstructBLIP (with Vicuna-7b language model). Both share similar CLIP ViT-L/14 vision encoders but differ in language models and fine-tuning objectives.

- **Cross-architecture transfer:** Perturbations are tested across fundamentally different model architectures: from BLIP-2 to Flamingo and vice versa. These models employ different vision encoders, language models, and architectural designs.

**Evaluation Setup:** We measure Attack Success Rate (ASR) for each vision-language task independently ($VQA_{general}$, $VQA_{specific}$, Classification, and Captioning). Success is measured by the target model failing to generate the correct output when given the perturbed image, regardless of input prompt.

### 5.4.2 Investigating Cross-Image Transferability via Augmentation Techniques

For evaluating universal adversarial perturbations that work across different images, we establish the following threat model:

**Adversary's Goal:** The attacker aims to generate a single perturbation pattern that, when applied to any image, causes a VLM to output a predetermined target text regardless of the input prompt or image content. This represents a higher level of attack efficiency compared to per-image optimization.

**Knowledge and Capabilities:** The attacker has access to a limited set of training images and their corresponding SCMix-augmented variants. The attacker has white-box access to the target model during perturbation optimization but must create a perturbation that generalizes to unseen images. The perturbation is restricted to an $L_\infty$ budget of $\epsilon = 16/255$.

**Evaluation Setup:** We evaluate universal perturbations on a separate set of test images not seen during training. Success is measured by the ASR across different task types. A successful attack causes the model to fail to generate the correct output text when presented with any previously unseen image, regardless of its content or the associated prompt.

Our SCMix-enhanced approach specifically aims to overcome the inherent limitations of gradient-based methods which tend to overfit to specific image features. By introducing controlled variations through self-mixing and cross-mixing during training, we hypothesize that the resulting perturbations will better capture model vulnerabilities independent of specific image characteristics.

### 5.4.3 Evaluation Rationale for Transferability experiments

In our experiments, we present all cross-prompt and cross-model transferability results under the untargeted attack setting. This decision is motivated by two primary factors. First, different Vision-Language Models (VLMs) incorporate visual information using distinct embedding mechanisms, making it difficult to pinpoint how input perturbations influence final output token embeddings. As observed, targeted transferability is notably higher between models sharing similar vision encoder architectures (e.g., BLIP-2 to InstructBLIP) but varies drastically otherwise, rendering direct comparison of targeted ASRs across heterogeneous architectures imprudent.

Second, the method of visual-textual fusion substantially impacts adversarial control. Decoder-only VLMs such as LLaVA and OpenFlamingo incorporate visual features late via cross-attention or token concatenation, treating them as auxiliary memory during causal language generation. Consequently, visual influence fades as decoding progresses (Awadalla et al. (2023), Liu et al. (2023).

In contrast, fusion-based models like BLIP-2 Li et al. (2023b) and InstructBLIP Dai et al. (2023) integrate visual features early and deeply into the input embeddings, ensuring persistent visual grounding throughout generation. Thus, targeted attacks are inherently more feasible in fusion-based models compared to decoder-only models (Dai et al. (2023)).

Given these architectural disparities, the evaluation of the transferability between models and between images through non-targeted ASR provides a more robust, fair, and comparable assessment across diverse VLM architectures.

Furthermore, we excluded CIA when comparing transferability as upon experimentation CIA yielded negligible transferability, and presented the following limitations for transferability fundamentally in the method itself:

- Image-Specific Semantic Embedding: CIA's perturbation strategy relies on per-image visual token manipulation to embed target concepts. This creates adversarial patterns that lack cross-image consistency and are dependent on the image since the gradient direction and token space for different images and concepts are vastly different and the token-space transformations between input/target image pairs are non-transferable.

- Model-Specific Embedding Architectures : CIA is highly dependent on the structure of visual and textual encoders, which fails to demonstrate cross-model transferability due to structural variations across LVLMs (BLIP2, LLaVA, Flamingo etc).

# 6 Results

As stated in Section 4, a major objective of our research is to reproduce the cross-prompt transferability results presented by Luo et al. (2024) for the CroPA attack. This section details our efforts to replicate those findings and provides a comparative analysis of our reproduced results against the original paper.

**Models Used:**

In reproducing the work of Luo et al. (2024), we evaluated five state-of-the-art Vision-Language Models (VLMs): OpenFlamingo, BLIP-2, InstructBLIP and LLaVA. For Flamingo, we utilized the open-source OpenFlamingo-9B implementation Awadalla et al. (2023), which provides comparable performance to the original model while being publicly accessible.

BLIP-2 introduces a two-stage fusion-based approach that first extracts visual features using a frozen CLIP image encoder, then processes these features through a Querying Transformer Li et al. (2023b). This architecture enables efficient adaptation to diverse vision-language tasks. The model employs OPT-2.7b as its language model component, facilitating flexible text generation capabilities.

InstructBLIP builds upon BLIP-2's fusion-based architecture while incorporating instruction tuning Dai et al. (2023). A key distinction is its use of the Vicuna-7b language model, which enhances the model's ability to follow task-specific instructions. This modification enables more precise control over the model's outputs through carefully crafted prompts.

LLaVA Liu et al. (2023) employs a decoder-only generation approach, where a pretrained CLIP ViT-L vision encoder connects to Vicuna-13B through a lightweight projection layer. Unlike fusion-based models, LLaVA passes visual features directly into the language model's context, treating images as special tokens in the prompt sequence.

These models represent two distinct architectural approaches: decoder-only models (OpenFlamingo, LLaVA) that treat visual embeddings as input context for generation, and fusion-based models (BLIP-2, Instruct-BLIP) that explicitly align visual and textual representations before generation. These differences significantly affect adversarial effectiveness across prompts in targetted attack.

Our reproduction efforts were based on the code provided in the authors' public repository, maintaining the original configurations of these models to ensure faithful comparison with the baseline results. While the overall implementation was well-documented, we encountered several challenges that required modifications for successful reproduction. Notably, the code for BLIP-2 and InstructBLIP models triggered multiple runtime errors, necessitating significant debugging and adjustments to achieve functional execution.

## 6.1 Claim 1: CroPA achieves cross-prompt transferability across various target texts. [Reproduced]

To validate the central claim of Luo et al. (2024), which posits that CroPA can achieve cross-prompt transferability, we meticulously replicated their experimental setup. This involved training adversarial examples using the CroPA method and evaluating their transferability across a diverse set of target prompts on the Flamingo VLM. Our evaluation focused on measuring the Targeted Attack Success Rate (ASR) across the following tasks: VQA (both general and specific), image classification, and image captioning.

### 6.1.1 Experimental Details

Following the experimental protocol outlined in Luo et al. (2024), we generated adversarial examples using CroPA and assessed their effectiveness against a range of target prompts distinct from those used during training. The specific target prompts used in our evaluation are listed in Table 1. These prompts were selected to represent a broad spectrum of semantic meanings, ensuring a rigorous evaluation of cross-prompt transferability. Additional details for all reproduced experiments are reported in Appendix D.1.

### 6.1.2 Comparative Analysis

Table 1 presents the Targeted ASRs achieved by the CroPA method on the Flamingo VLM for different target texts. Overall, our results align well with the claim that CroPA can achieve cross-prompt transferability. We observed non-zero ASRs across all evaluated tasks, indicating that adversarial examples generated using CroPA were effective in misleading the VLM, even when presented with diverse target prompts. However, we do note that CroPA consistently under-performs for tasks such as Classification and Captioning, especially with texts such as "too late" or "not sure" that have abstract meanings and don't have significant visual semantics.

| Target Prompt | Method | VQA$_{general}$ | VQA$_{specific}$ | Classification | Captioning | Overall |
|---|---|---|---|---|---|---|
| unknown | Single-P | 0.2640 | 0.4720 | 0.2700 | 0.1400 | 0.2865 |
| | Multi-P | 0.7240 | 0.8740 | 0.5550 | 0.2850 | 0.6095 |
| | CroPA | **0.9680** | **0.9880** | **0.7070** | **0.4200** | **0.7708** |
| I am sorry | Single-P | 0.2120 | 0.4740 | 0.5170 | 0.2530 | 0.3640 |
| | Multi-P | 0.7020 | 0.9020 | 0.7090 | 0.6140 | 0.7318 |
| | CroPA | **0.8620** | **0.9260** | **0.7170** | **0.6890** | **0.7985** |
| not sure | Single-P | 0.3540 | 0.4240 | 0.1450 | 0.0000 | 0.2308 |
| | Multi-P | 0.6520 | 0.6860 | 0.2310 | **0.0030** | 0.3930 |
| | CroPA | **0.8760** | **0.8940** | **0.2420** | 0.0010 | **0.5033** |
| very good | Single-P | 0.3980 | 0.5760 | 0.2180 | 0.0480 | 0.3100 |
| | Multi-P | 0.8940 | 0.9640 | 0.4040 | 0.2080 | 0.6175 |
| | CroPA | **0.9620** | **0.9860** | **0.6060** | **0.2540** | **0.7020** |
| too late | Single-P | 0.2520 | 0.4800 | 0.2280 | 0.0220 | 0.2455 |
| | Multi-P | 0.7880 | 0.9120 | 0.5040 | 0.1630 | 0.5918 |
| | CroPA | **0.9300** | **0.9580** | **0.7010** | **0.1790** | **0.6920** |
| metaphor | Single-P | 0.3280 | 0.6020 | 0.5550 | 0.1040 | 0.3973 |
| | Multi-P | 0.8880 | 0.9520 | 0.8420 | 0.4910 | 0.7933 |
| | CroPA | **0.9840** | **0.9940** | **0.9100** | **0.5840** | **0.8680** |

Table 1: Targeted ASRs tested on Flamingo with different target texts, across a variety of Vision-Language tasks. Here VQA$_{general}$ and VQA$_{specific}$, refer to the degree of specificity of prompts pertaining to a particular image. The prompts used are reported in Appendix E

Our experiments successfully reproduce the core finding that CroPA exhibits robust cross-prompt transferability, thereby validating the adversarial vulnerability of VLMs to such attacks.

### 6.2 Claim 2: CroPA achieves the best overall performance across different number of prompts [Reproduced]

We conduct experiments to compare the performance of baseline and CroPA methods over different number of prompts compared to baselines as shown in Figure 4. For all methods ASR increases with increase in number of prompts (redundant for Single-P as it inherently gets tested on a single prompt, hence the lone "star" symbol in Figure 4). We note that CroPA gives better ASR for the same number of prompts as compared to baselines, and is a consistent better performer, underscoring the core motivation of transferability across prompts. We select the target text "unknown" to avoid the inclusion of high-frequency responses commonly found in vision-language tasks, which we find a valid argument presented by Luo et al. (2024), for this experiment.

We can also observe that the baseline method saturates in an ASR rate lower than that of CroPA as number of prompts are increased, indicating a trend that by adding more prompts, the baseline approach cannot surpass the performance of CroPA methods, which is consistent with the findings of Luo et al. (2024).

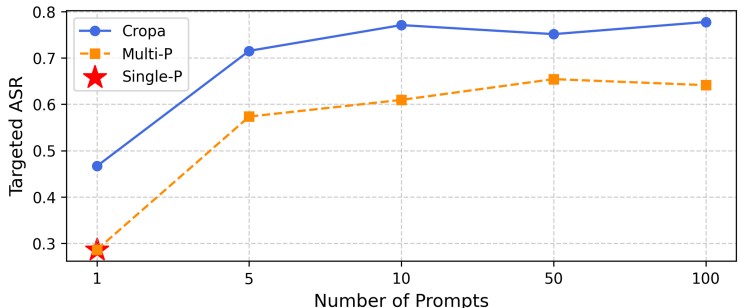

Figure 4: Targeted ASR tested with different number of prompts on Flamingo. With the same number of attack iterations, our CroPA significantly outperforms baselines. "unknown" is used as the target text.

### 6.3   Claim 3: CroPA converges towards higher ASR as number of iterations increase [Reproduced]

Our obtained results on investigating the converging ASR rate for CroPA resulted in contrasting results with respect to the original reported metrics of convergence with higher number of iterations (refer Appendix A.2), for the author's results, despite using the exact same parameters for training and evaluation, as well as the author's own code. Where the original metrics showed CroPA converging with higher number of prompts. While this certainly is the trend observed with the overall ASR across tasks as shown in Figure 6, which imitates the figures the original paper reports, when we investigated the ASR convergence for each task's use case, individually the trend is prone to instability. This is verified by the findings reported in 5. Additionally, the model used for evaluation under this scenario wasn't specified by the authors, hence we opted to report figures obtained for the OpenFlamingo model with the target text being "unknown".

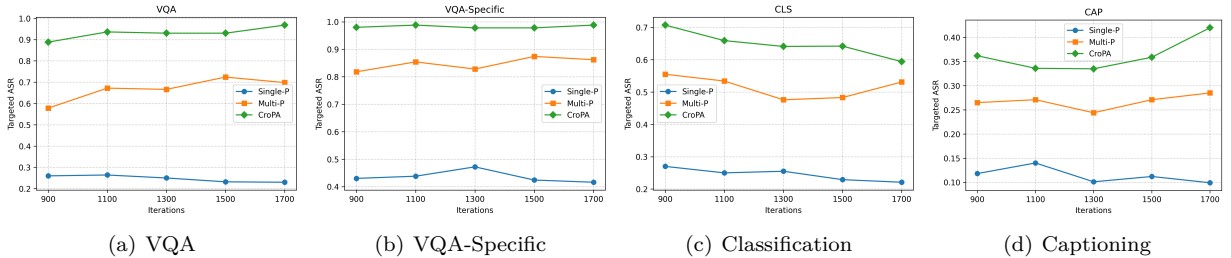

(a) VQA             (b) VQA-Specific             (c) Classification             (d) Captioning

Figure 5: Comparison of Single-P, Multi-P, and CroPA across different categories.

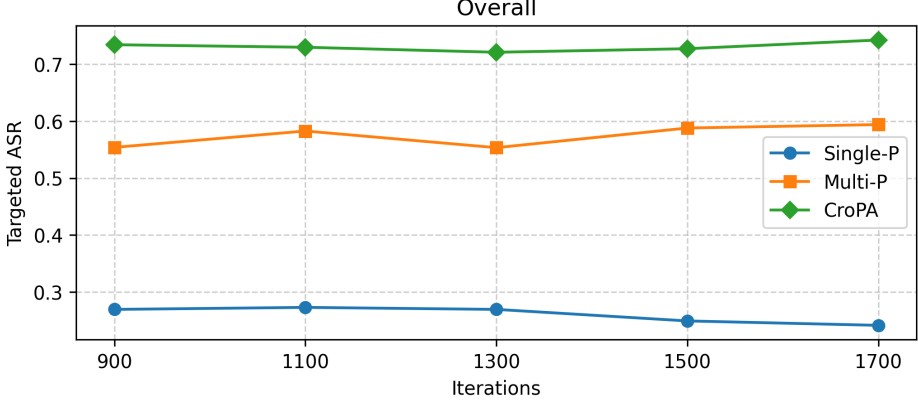

Figure 6: Overall Convergence and Comparison of CroPA

### 6.4 Claim 4: CroPA outperforms baselines under few-shot settings [Reproduced]

In addition to the textual prompt, the Flamingo model also supports providing extra images as in-context learning examples to improve the task adaptation ability. To investigate their effect on the cross-prompt attack, Luo et al. (2024) tested the ASRs of the image adversarial examples with the number of in-context learning examples different from the one provided in the optimisation stage. During the optimisation stage, the image adversarial examples are updated under the 0-shot setting, namely no extra images are provided as the in-context learning examples. In the evaluations, the 2-shot setting is used, i.e. two extra images are used as the in-context learning examples.

| Setting | Method | $VQA_{general}$ | $VQA_{specific}$ | Classification | Captioning | Overall |
|---------|--------|------------|-------------|----------------|------------|---------|
| shot=0 | Multi-P | 0.7240 | 0.8740 | 0.5550 | 0.2850 | 0.6095 |
| | CroPA | **0.9680** | **0.9880** | **0.7070** | **0.4200** | **0.7708** |
| shot=2 | Multi-P | 0.7860 | 0.9420 | 0.5630 | 0.0010 | 0.5730 |
| | CroPA | **0.9440** | **0.9940** | **0.6860** | **0.0300** | **0.6635** |

Table 2: Targeted ASRs with and without visual in-context learning. The shot indicates the number of images added for in-context learning.

The use of in-context examples makes it inherently harder for adversarial attacks to succeed as they can cause a shift from the generation conditions from the optimization stage (Luo et al. (2024)). Furthermore, adding in-context images makes the model more contextually aware of the correct prompt space which diverges from the target text space. A number of works (Zhao et al. (2023),Qi et al. (2023)), verify that attacks like CroPA which are optimized for specific context lose effectiveness when the context changes.

However, it is worthy to note that CroPA performs significantly better than the baseline in most tasks, making it more robust to widening of the context space by in-context examples. This verifies the motivation and hypothesis behind min-max objective detailed in Subsection 4.3, which was formulated to make CroPA more robust to widening of the prompt-space and consequent changes in context.

### 6.5 Results Beyond the Original Paper

The following subsections detail our extended experiments building upon CroPA's framework and incorporating novel enhancements, mitigating the corresponding limitations of CroPA.

#### 6.5.1 Noise Initialization

Our proposed Noise Initialization via Vision Encoding Optimization (detailed in subsection 5.1) demonstrates significant improvements across various tasks compared to baseline methods. By using semantically informed perturbation initialization, we achieve superior performance in adversarial robustness and cross-prompt transferability. As shown in Table 3, the integration of our method (CroPA+Init) consistently outperforms both CroPA and Multi-P across all metrics, achieving 6-15% gains in overall accuracy.

These results validate the efficacy of semantically guided initialization over random noise approaches. The vision-encoder-anchored perturbations not only enhance attack success rates but also improve efficiency by reducing optimization steps. This highlights the robustness and adaptability of our method as a principled alternative for adversarial attack initialization.

**Theoretical Analysis of Initialization in Bi-level Optimization** The significant performance gains achieved through our diffusion-guided initialization can be attributed to the unique optimization challenges in CroPA's min-max framework. Our analysis reveals three key factors:

First, CroPA's adversarial formulation resembles a saddle-point problem where optimization stability is heavily influenced by initialization. Zhang et al. (2025) demonstrated that min-max optimization problems exhibit significantly higher sensitivity to initialization conditions than standard minimization problems, often requiring carefully designed initialization strategies to achieve stable convergence. This sensitivity

| Target Prompt | Method | VQA$_{general}$ | VQA$_{specific}$ | Classification | Captioning | Overall |
|---|---|---|---|---|---|---|
| | Multi-P | 0.7240 | 0.8740 | 0.5550 | 0.2850 | 0.6095 |
| unknown | CroPA | **0.9680** | **0.9880** | 0.7070 | 0.4200 | 0.7708 |
| | CroPA+Init | 0.8824 | 0.8920 | **0.8858** | **0.9306** | **0.8977** |
| | Multi-P | 0.6840 | 0.8260 | 0.9050 | 0.6090 | 0.7560 |
| bomb | CroPA | 0.8176 | 0.9100 | **0.9498** | 0.6960 | 0.8433 |
| | CroPA+Init | **0.9140** | **0.9468** | 0.9020 | **0.9412** | **0.9260** |
| | Multi-P | 0.7020 | 0.9020 | 0.7090 | 0.6140 | 0.7318 |
| I am sorry | CroPA | 0.8620 | 0.9260 | 0.7170 | 0.6890 | 0.7985 |
| | CroPA+Init | **0.9320** | **0.9524** | **0.9270** | **0.9716** | **0.9458** |
| | Multi-P | 0.7900 | 0.8860 | 0.6700 | 0.6700 | 0.7540 |
| very good | CroPA | 0.8540 | 0.8990 | 0.7680 | 0.7550 | 0.8190 |
| | CroPA+Init | **0.8745** | **0.8894** | **0.8362** | **0.9043** | **0.8761** |
| | Multi-P | 0.6960 | 0.8540 | 0.6780 | 0.6060 | 0.7585 |
| too late | CroPA | 0.8990 | 0.9060 | 0.7940 | 0.8150 | 0.8535 |
| | CroPA+Init | **0.9340** | **0.9220** | **0.9110** | **0.9660** | **0.9333** |

Table 3: Targeted ASRs on Blip2 with different target texts using CroPA Vision Encoder noise initialization. The detailed hyperparameters used are discussed under section D.2.

becomes particularly pronounced in CroPA's architecture, where prompt perturbations continuously alter the optimization landscape throughout training.

Second, the alternating update schedule in CroPA—which updates image perturbations more frequently than prompt perturbations—creates optimization asymmetry that further amplifies initialization importance. Chen et al. (2019) identified that such alternating gradient-based methods in min-max problems frequently suffer from local cycling behaviors, with convergence properties strongly dependent on the initial point's proximity to adversarially robust regions. Our semantic initialization leverages this insight by deliberately placing the starting point closer to meaningful adversarial directions.

Third, empirical evidence from our experiments aligns with findings from Wu et al. (2020), who demonstrated that perturbations initialized with semantic priors consistently transfer better across models than randomly initialized ones, producing substantial improvements in attack success rates. This effect becomes particularly pronounced in bi-level optimization contexts like CroPA, where optimization pathways are more complex.

Our diffusion-based approach addresses these challenges by initializing perturbations in the direction of semantic targets rather than using standard random noise. This principled initialization strategy enables faster convergence during optimization while simultaneously enhancing the transferability of the resulting adversarial examples. The significant performance improvements observed in Table 3, which show significant ASR increases.

| Method | VQA$_{general}$ | VQA$_{specific}$ | Classification | Captioning | Overall |
|---|---|---|---|---|---|
| Multi-P | 0.7053 | 0.8807 | 0.7001 | 0.5553 | 0.7104 |
| CroPA | 0.8723 | 0.9457 | 0.7518 | 0.6138 | 0.7958 |
| CIA | 0.2985 | 0.2281 | 0.4857 | 0.4687 | 0.370 |
| CroPA+Init | **0.9145** | **0.9342** | **0.8982** | **0.9464** | **0.9209** |

Table 4: Average Targeted ASRs on BLIP-2 for different methods. Results show the impact on different vision-language tasks: Visual Question Answering (general and specific), Classification, and Captioning. The best performance values for each task are highlighted in bold. Apart from the previously mentioned baselines, we also compare our results with the Contextual Injection Attack (CIA) Yang et al. (2024) as an additional baseline.

Gradient based adversarial attacks often take up semantics of their targets, which theoretically shouldn't need a semantically aligned initialization at all, however, due to the min-max complexities we mention above, the semantic alignment produced by gradients weaken, and injection of target semantics into initialization is

necessary to strengthen them and aid better convergence. To aid this claim we provide experimental results in Figure 7.

We evaluate the incorporation of target attributes into the perturbations by first taking the CLIP score of the target text and our noise perturbations obtained by the base CroPA method and our Init enhancement. This offers an insight as to how much of the target features are taken up by the perturbations.

It is observed that CroPA does show expected behavior of gradient-based perturbation methods and acquires the context of the target, however we do notice a modest increment in the CLIP score with the Initialization included. This suggests that while the target semantics are acquired by CroPA's perturbations, the initialization provides a subtle change in favor of better proximity it adversarial vulerable regions, even if it does not significantly upgrade the semantic features acquired by the perturbations with respect to the target.

The convergence trend before and after the updated initialization also shows a promising trend aligned with our original hypothesis that, by deliberately placing the starting point closer to meaningful adversarial directions, our semantic initialization strategy achieves faster convergence.

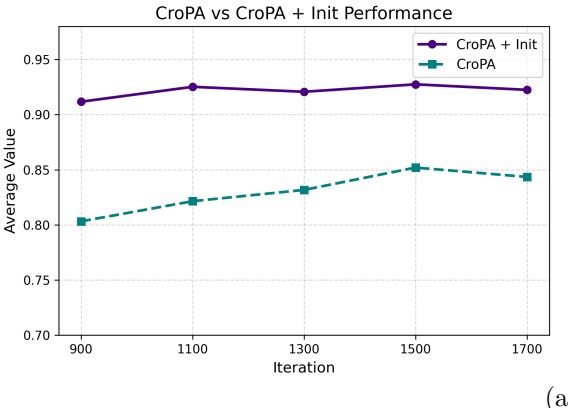

| Method | CLIP Score |
|---|---|
| CroPA | 21.80 |
| CroPA+Init | **23.09** |

Target Text: "Bomb"

(a)                                                (b)

Figure 7: (a) Overall Convergence Comparison of base CroPA and CroPA+Init. (b) Quantifying feature similarity of learned perturbation to target by CLIP score.

### 6.5.2 Investigating Cross-Image transferability and Image Augmentation

CroPA's effectiveness stems from its ability to retain adversarial impact across different textual prompts, but this focus inherently limits its cross-image transferability. CroPA optimizes perturbations specifically to be effective across different prompts, the resulting adversarial signals remain tightly coupled to the low-level features of the image they were generated on. This strong dependency prevents the perturbations from generalizing to different images, making universal cross-image attacks ineffective. This weakens the author's claims that learning the perturbation across images would lead to cross-image transferability.

To address this limitation, we further explored the integration of the SCMix and CutMix frameworks, presented in subsection 5.3, both of which introduce input diversity by mixing image features during training. The goal was to encourage perturbations to extend beyond a single image and improve their generalization. We believe CutMix outperforms SCMix in promoting adversarial transferability because its region-based mixing maintains distinct, non-overlapping spatial areas. This encourages the model to learn modular and independently manipulable features, which in turn makes adversarial perturbations more generalizable across different inputs. Our results (Table 5) show that while cross-image transferability remains a challenge, both SCMix and CutMix contribute to measurable improvements. Despite these gains, the performance between cross-image methods and CroPA when applied to a single image showcase a fundamental limitation of CroPA: it excels at cross-prompt robustness but does not effectively support transferable adversarial attacks across images.

| Method | VQA | VQA$_{specific}$ | Classification | Captioning | Overall |
|---|---|---|---|---|---|
| CroPA (Base) | 0.3500 | 0.4920 | 0.4000 | 0.5000 | 0.4355 |
| CroPA with SCMix | **0.6560** | 0.7160 | **0.5000** | **0.9500** | **0.7055** |
| CroPA with CutMix | 0.4520 | **0.7240** | **0.5000** | 0.5500 | 0.5565 |

Table 5: Untargeted ASR of CroPA in cross-image settings with and without SCMix and CutMix. The results represent the best ASRs achieved across all iterations. CroPA without input mixing attained its highest ASRs at 1700 iterations, while CroPA with SCMix and CutMix outperformed it significantly earlier, at just 1500 iterations and 1600 iterations, respectively. Additional details on experimental parameters are reported in Appendix D.3.

### 6.5.3 Guiding perturbations via Target Value Vectors

We obtain results for the D-UAP-modified CroPA framework detailed in Subsection 5.2. Our results in Tables 6 and 7 demonstrate the effectiveness of the CroPA+D-UAP method across various vision-language tasks. The enhanced perturbations greatly improve upon the existing algorithm across the board to give an overall ASR of 96.85% which is an 12.5% increase from CroPA's base method. The enhanced algorithm particularly improves ASR in tasks such as VQA$_{general}$ and Captioning, marking a 13% and 28% increase respectively. "Bomb" was used as the target text in these experiments, which underscores the performance of the improved method even with instances of harmful text which inherently are set to trigger a model's guardrails. These means that for the worse expected cases of "harmful words" the enhancement lead the model to perform significantly better in vulnerable scenarios.

| Method | VQA$_{general}$ | VQA$_{specific}$ | Classification | Captioning | Overall |
|---|---|---|---|---|---|
| Multi-P | 0.68 | 0.82 | 0.90 | 0.60 | 0.75 |
| CroPA | 0.8176 | 0.9100 | 0.9498 | 0.6960 | 0.8433 |
| CIA | 0.3431 | 0.3102 | 0.4732 | 0.5534 | 0.4199 |
| CroPA+D-UAP | **0.9420** | **0.9720** | **0.9790** | **0.9810** | **0.9685** |

Table 6: Targeted ASRs on Blip2 using D-UAP function and target text being "Bomb", outperforming PGD baselines and CIA, as well as the base CroPA method.

| Method | VQA$_{general}$ | VQA$_{specific}$ | Classification | Captioning | Overall |
|---|---|---|---|---|---|
| Multi-P | 0.6960 | 0.8540 | 0.8990 | 0.6060 | 0.7638 |
| CroPA | 0.7900 | 0.8860 | 0.9470 | 0.6700 | 0.8233 |
| CIA | 0.3027 | 0.4302 | 0.5112 | 0.5080 | 0.4380 |
| CroPA+D-UAP | **0.9000** | **0.9520** | **0.9500** | **0.9060** | **0.9270** |

Table 7: Targeted ASRs on Open Flamingo using D-UAP Loss function and target text being "Bomb".

### 6.5.4 Analyzing Cross Model Transferability by Value Vector guidance

The method also presents a strong case for increasing cross-model transferability as reported in Table 8, with significant improvements over the vanilla CroPA method, especially in Classification and Captioning tasks (14% and 21% respectively), with modest increase VQA. The discrepancy across tasks can be attributed to the fact that VQA prompts span a much more complex embedding space than targeted tasks like Classification or Captioning where the output is limited to much smaller spans.

It's important to address a key limitation we observed during our analysis, particularly concerning the vision encoder's behavior, which verifies our hypothesis in Subsection 5.2 about the method's limitations.

A limitation of the CroPA Doubly-UAP approach lies in the dependency of value vectors on the specific vision encoder used within the Vision Language Model (VLM). This can limit transferability to model's having the same or similar architectural design of the Vision Encoder, this is verified in Table 8, where the

pertubrations trained on BLIP2 appear to be passably transferable to InstructBLIP which used the same Vision Encoder, and failing to transfer to OpenFlamingo. This expected pattern can verified also in the case where the perturbations are trained on Flamingo and applied to the other models, where the ASR comes to be insignificant (refer Table D.4.5 in Appendix D.4.5. Experimental details are reported in Appendix D.4.

| Settings | Method | VQA$_{general}$ | VQA$_{specific}$ | Cls | Cap | Overall |
|---|---|---|---|---|---|---|
| BLIP2 to InstructBLIP | Multi-P | 0.1760 | 0.2090 | 0.2750 | 0.2480 | 0.2270 |
| | CroPA | 0.3220 | 0.4170 | 0.6720 | 0.6150 | 0.5065 |
| | CroPA + D-UAP | **0.3940** | **0.5780** | **0.8800** | **0.8880** | **0.6850** |
| BLIP2 to Flamingo | Multi-P | 0.0810 | 0.0660 | 0.0760 | 0.0890 | 0.0780 |
| | CroPA | 0.1520 | 0.1220 | 0.1180 | 0.1820 | 0.1435 |
| | CroPA + D-UAP | **0.1850** | **0.2190** | **0.1420** | **0.3880** | **0.2335** |
| BLIP2 to LLaVA | Multi-P | 0.1220 | 0.1060 | 0.0810 | 0.0630 | 0.0930 |
| | CroPA | 0.3190 | 0.2570 | 0.2730 | 0.3670 | 0.3040 |
| | CroPA + D-UAP | **0.5070** | **0.3420** | **0.4380** | **0.8430** | **0.5325** |
| InstructBLIP to BLIP2 | Multi-P | 0.1230 | 0.1520 | 0.1020 | 0.0960 | 0.1183 |
| | CroPA | 0.2690 | 0.2320 | 0.1870 | 0.2270 | 0.2288 |
| | CroPA+D-UAP | **0.4800** | **0.5100** | **0.4700** | **0.5000** | **0.4900** |
| InstructBLIP to LLaVA | Multi-P | 0.0650 | 0.0580 | 0.0420 | 0.0370 | 0.0505 |
| | CroPA | 0.1620 | 0.1120 | 0.1410 | 0.1230 | 0.1345 |
| | CroPA+D-UAP | **0.3450** | **0.3150** | **0.3680** | **0.3950** | **0.3558** |
| InstructBLIP to Flamingo | Multi-P | 0.0320 | 0.0210 | 0.0120 | 0.0110 | 0.0190 |
| | CroPA | 0.0740 | 0.0680 | 0.0490 | 0.0480 | 0.0598 |
| | CroPA+D-UAP | **0.1600** | **0.1400** | **0.1200** | **0.1300** | **0.1375** |
| Flamingo to BLIP2 | Multi-P | 0.0140 | 0.0070 | 0.0090 | 0.0120 | 0.0105 |
| | CroPA | 0.0450 | 0.0210 | 0.0300 | 0.0380 | 0.0335 |
| | CroPA+D-UAP | **0.1080** | **0.0800** | **0.0990** | **0.1180** | **0.1013** |
| Flamingo to InstructBLIP | Multi-P | 0.0100 | 0.0060 | 0.0050 | 0.0090 | 0.0075 |
| | CroPA | 0.0360 | 0.0220 | 0.0280 | 0.0300 | 0.0290 |
| | CroPA+D-UAP | **0.1050** | **0.0720** | **0.0880** | **0.1100** | **0.0938** |
| Flamingo to LLaVA | Multi-P | 0.0180 | 0.0110 | 0.0140 | 0.0170 | 0.0150 |
| | CroPA | 0.0400 | 0.0300 | 0.0350 | 0.0430 | 0.0370 |
| | CroPA+D-UAP | **0.1100** | **0.0850** | **0.0980** | **0.1170** | **0.1025** |

Table 8: The cross model test under different settings. "BLIP2 to InstructBLIP" means the perturbations optimised on BLIP2 are tested on the InstructBLIP and similarly for "BLIP2 to Flamingo"

Our investigations revealed that for a given target text, the value vectors generated by the vision encoder remain approximately consistent across different input images. This observation has several implications:

1. **Vision Encoder Specificity**: The value vectors produced are aligned to the architecture and pre-training of the vision encoder.

2. **Limited Diversity in Perturbations**: Due to the consistency of value vectors for a given target text, the diversity of adversarial perturbations generated by our method may be constrained. This could potentially limit the attack's ability to produce highly varied adversarial examples.

3. **Target Text Dependency**: The similarity in value vectors across different images for the same target text indicates that the attack's outcome is strongly influenced by the choice of target text rather than the specific features of the input image.

# 7 Discussion

## 7.1 Observations with reproducibility

Our study aimed to replicate key aspects and findings of Luo et al. (2024) on the Cross-Prompt Attack (CroPA) framework, we have substantiated the original claims regarding the enhancement of cross-prompt adversarial transferability in Vision-Language Models (VLMs). Our experiments confirmed that CroPA achieves strong cross-prompt transferability across diverse tasks, including VQA, image classification, and captioning, consistently outperforming baseline methods. CroPA remains robust in few-shot settings (6.4), outperforming other baselines and demonstrating resilience to the shift from the generation conditions from the optimization stage introduced by in-context learning examples.

## 7.2 Noise Initialization findings

In Subsection 5.1, we identified a critical limitation in CroPA and related adversarial attack methods, the reliance on random noise initialization, which is agnostic to the semantic structure of the target prompt. We hypothesized that this lack of semantic guidance at initialization introduces instability, prolongs convergence, and reduces cross-prompt and cross-modal transferability.

To address this, we proposed a semantically informed initialization strategy using a diffusion-synthesized reference image and alignment in the vision encoder's feature space, designed to orient perturbations in a meaningful adversarial direction from the outset. Our experiment results in 6.5.1 empirically validate this hypothesis: across all evaluated benchmarks, our noise initialization method significantly outperformed both CroPA and random-initialized baselines, yielding up to 15% gains in Targeted ASR and notably reducing optimization steps. These findings confirm that semantically anchored initialization not only enhances adversarial efficacy but also provides a principled and efficient foundation for VLM attack frameworks.

This enhancement not only increased the effectiveness of adversarial attacks but also reduced the number of optimization steps required, making the process more efficient. By anchoring perturbations to the vision encoder's semantic space, this method provided a more principled and robust initialization strategy compared to random noise, highlighting its potential for improving adversarial attack frameworks.

## 7.3 Evaluation of Transferability Experiments

### 7.3.1 Cross-Image Transferability

Despite the author's claims that and that cross-image transferability can be achieved by computing perturbations across image, we found in our experimentation that while CroPA excels at cross-prompt transferability, its adversarial perturbations are tightly coupled to the low-level features of the specific image they were generated for. To address CroPA's inherent limitation in cross-image transferability, we integrated SCMix (Subsection 5.3) and CutMix, both image augmentation methods that help increase cross-image diversity while preserving semantic information of the original images.

Our results 6.5.2 showed that SCMix provided significant improvements in cross-image transferability, particularly in tasks like VQA, VQA-specific and Captioning, where the ASR increased overall by approximately 27 percent. CutMix, on the other hand, achieved a net increase of around 12 percent over the base cross-image version of CroPA. We believe this is because SCMix explicitly introduces spatial and scale invariance through its random cropping and resizing of the same image, then softly mixes it with a second image. This encourages the perturbations to target semantically aligned features that are less sensitive to exact positioning or scale, making it easier for adversarial perturbations to generalize across different inputs.

We can see that adding SCMix and CutMix improves the success of cross-image transferability when applied to a cross-prompt transferable framework like CroPA. Both our improvements to address cross-image transferability are image augmentation techniques, which we believe are able to add cross-image features for better generalization when learning the perturbations. For further improvements, future research aimed at improving cross-image transferability in cross-prompt frameworks should focus on more complex and layered

approaches beyond augmentation such as combining it with feature-alignment attacks based on the latent representation of the images, or perturbations in the latent space directly among other approaches.

### 7.3.2   Cross-Model Transferability

The D-UAP-modified CroPA framework (Subsection 5.2 represents a significant advancement in CroPA's attack method, achieving an overall Targeted Attack Success Rate (ASR) of 96.85%, an 13.5% improvement over the base CroPA method as shown in 6.5.3. This enhancement is particularly notable in tasks like VQA$_{general}$ and Captioning, where ASRs increased by 13% and 28%, respectively. These improvements were observed even when targeting challenging or harmful text, such as "Bomb," which typically triggers model guardrails, demonstrating the robustness of the D-UAP framework in bypassing safety mechanisms.

Additionally, the framework exhibited improved cross-model transferability, especially between models with similar vision encoders, such as BLIP2 and InstructBLIP. However, the method's effectiveness is constrained by its dependency on the specific vision encoder, which limits the diversity of perturbations and hinders transferability to models with different architectures.

This limitation underscores the broader challenge of developing encoder-agnostic adversarial methods. Despite this, the D-UAP framework marks a critical step forward in creating universal adversarial perturbations that are effective across both prompts and images, paving the way for future research to enhance generalizability and robustness in vision-language models. Lastly, our cross-model transferability comparisons do not include evaluation with the Contextual Injection Attack, since the attack is inherently non-transferable and was unable to produce comparable results.

## 7.4   Trade-offs Between Different Types of Adversarial Transferability

Our comprehensive analysis reveals fundamental trade-offs between different dimensions of adversarial transferability in vision-language models. These trade-offs are essential to understand when designing robust adversarial attacks:

### 7.4.1   Cross-Prompt vs. Cross-Model Transferability

The original CroPA method excels at cross-prompt transferability by optimizing perturbations against competing text embeddings. However, this optimization implicitly anchors the perturbations to the specific architecture of the target model. Our experiments in Section A.1 confirm this limitation, with ASRs dropping significantly when perturbations generated on one model are applied to another.

Our D-UAP enhancement (Section 5.2) attempts to address this trade-off by targeting generalizable components within vision encoders rather than model-specific features. While this improves cross-model transferability between architecturally similar models (e.g., BLIP-2 and InstructBLIP), the improvement remains modest between fundamentally different architectures (e.g., BLIP-2 and Flamingo). This suggests an inherent tension between optimizing for prompt diversity and model diversity simultaneously.

### 7.4.2   Cross-Prompt vs. Cross-Image Transferability

Cross-prompt and cross-image transferability represent competing optimization objectives. CroPA achieves strong cross-prompt transferability by tightly coupling perturbations to image-specific features. This coupling, however, fundamentally limits cross-image generalization.

We propose using data augmentation to address this limitation. Our SCMix and CutMix integrations (Section 5.3) attempt to overcome this limitation by diversifying the optimization process through augmentation. The improvements observed in Table 5 are promising for future work in this direction. However, at the same time, we find it important to point out that our findings suggest that while augmentation techniques can improve cross-image transferability, they do not fully resolve the fundamental tension between these objectives.

### 7.4.3 Role of Initialization in Transferability Balance

Our diffusion-based initialization approach (Section 5.1) represents an alternative perspective on these trade-offs. By anchoring perturbations to semantically meaningful initializations, we can achieve higher ASRs without sacrificing cross-prompt transferability. This suggests that the optimization landscape's starting point significantly influences the achievable balance between different transferability types.

Unlike methods that modify the optimization process itself, better initialization enables perturbations to converge toward more general adversarial directions that remain effective across prompts. However, even with improved initialization, the fundamental tension between cross-prompt and cross-image/cross-model transferability remains. A perturbation optimized for one dimension inevitably sacrifices effectiveness in others.

### 7.4.4 Implications for Future Research

These observed trade-offs highlight a critical challenge in adversarial machine learning for vision-language models: no single attack methodology can simultaneously maximize all forms of transferability. Future research should focus on developing frameworks that allow for controlled balance between these competing objectives, potentially through multi-objective optimization approaches or architecturally-aware attack designs.

Our work demonstrates that while individual techniques can push the boundaries of specific transferability dimensions, a comprehensive solution would require fundamentally rethinking how adversarial perturbations interact with the underlying model architectures and data representations.

Despite these affirmations, the domain of cross-prompt adversarial transferability remains underexplored. The considerable lack of literature in this area highlights a gap in our understanding of prompt-induced vulnerabilities within VLMs. Addressing this gap is imperative, as it holds profound implications for the secure deployment of these models.

### 7.5 Broader Impact

From an attacker standpoint, adversarial examples exhibiting high cross-prompt transferability pose significant threats. Such examples can manipulate VLMs to produce malicious or misleading outputs, even when prompted with harmless queries. This capability could be exploited to disseminate false information or to subvert systems dependent on VLMs for content generation and decision-making.

Conversely, from a defensive perspective, the application of imperceptible perturbations offers a novel mechanism to safeguard sensitive information. By embedding these perturbations into images, it is possible to induce VLMs to consistently output predetermined, non-sensitive text, thereby thwarting unauthorized attempts to extract confidential data from personal images. This technique serves as a proactive measure to enhance privacy and data security in an era where visual data is increasingly susceptible to exploitation.

Our study also introduces refinements to the CroPA framework, including improved initialization strategies and an enhanced loss function. These modifications have demonstrated a marked increase in both the Attack Success Rate (ASR) and the generalizability of adversarial examples across different models and images. Such advancements not only reinforce the efficacy of cross-prompt attacks but also pave the way for more resilient defenses against them.

In conclusion, while our reproducibility study affirms the foundational work of Luo et al. (2024), it also accentuates the necessity for deeper investigation into cross-prompt adversarial transferability. A comprehensive understanding of this phenomenon is crucial for developing robust VLMs capable of withstanding adversarial manipulations and for formulating effective countermeasures to protect user data and maintain the integrity of model outputs.

### 7.6 Limitations and Future Work

While we were successfully able to reproduce the key results of CroPA and conduct extensive ablations, our progress was hindered on multiple occasions due to a lack of computational resources. Training and evaluating adversarial attacks on large-scale VLMS, such as Flamingo and BLIP2, require significant GPU memory and processing power, which limited the scale and depth of some experiments. Running extensive hyperparameter tuning or evaluating cross-model transferability across a wider range of architectures was often constrained by resource availability. Additionally, we aimed to further investigate transferability using an ensemble of these models, which would have provided deeper insights into the generalizability of adversarial perturbations. However, this approach required very high memory and compute requirements, making it infeasible with our current infrastructure

Future work could focus on improving further cross-model transferability, particularly by developing encoder-agnostic methods as well as enhancing cross-image generalization. Another way to further improve the practical applicability of the method is to implement the optimization with query-based strategies.

### 7.7 Communication With Original Authors

No direct communication could be established with the original authors during the replication process. The issue raised on their GitHub repository regarding the BLIP-2 and InstructBLIP models remains unresolved at the time of writing.

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

# Appendix

# Table of Contents for the Paper

# A  Results Sourced from the original paper

## A.1  Cross-Model Transferability Evaluation for CroPA

Table 9 presents CroPA's cross model transferability results, underscoring the model-specific over-fitting issue highlighted in 5.3.

| Settings | Method | VQA$_{general}$ | VQA$_{specific}$ | Classification | Captioning | Overall |
|---|---|---|---|---|---|---|
| BLIP2 to InstructBLIP | Multi-P | 0.00 | 0.01 | 0.04 | 0.03 | 0.02 |
| | CroPA | **0.00** | **0.04** | **0.15** | **0.11** | **0.08** |
| InstructBLIP to BLIP2 | Multi-P | 0.00 | 0.02 | 0.10 | 0.02 | 0.04 |
| | CroPA | **0.01** | **0.05** | **0.13** | **0.04** | **0.06** |

Table 9: The cross model test under different settings. "BLIP2 to InstructBLIP" means the perturbations optimised on BLIP2 are tested on the InstructBLIP and "InstructBLIP to BLIP2" are different. The language model for BLIP2 is OPT-2.7b and the model for InstructBLIP is Vicuna-7b. The best performance values for each task are highlighted in bold.

## A.2  Convergence results of the original paper

Figure 8 presents CroPA's convergence results over increasing number of iterations, referred in 6.3.

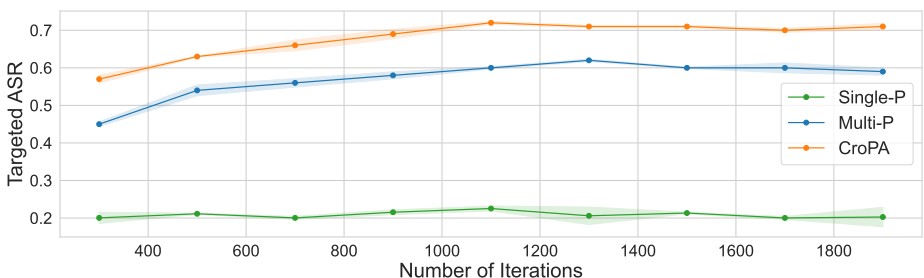

Figure 8: Overall Convergence and Comparison of Original CroPA

# B  Additional Method Details

In this section we present the algorithm formally depicted for our enhancement methodologies to CroPA as discussed in subsection 5.1 and 5.2 in Algorithms 2 and 3 respectively :

The modified algorithm to apply SCMix image augmentation with CroPA is presented in Algorithm 4.

---

**Algorithm 2** Diffusion-Based Noise Initialization and PGD

**Require:** $x$, $T$, $f_v$, $\epsilon$, $\alpha$, $\theta_{SDXL}$

  — **Noise Initialization** —

  Sample noise: $z \sim \mathcal{N}(0, I)$

  Generate target image: $x_{target} \leftarrow D(T, z; \theta_{SDXL})$

  Compute initial perturbation:

  $\delta_{init} \leftarrow \arg\min_{\|\delta\|_\infty \leq \epsilon} \|f_v(x + \delta) - f_v(x_{target})\|_2^2$

  — **Adversarial Optimization via PGD** —

  Initialize adversarial example: $x_{adv} \leftarrow x + \delta_{init}$

  **for** $t = 1$ to $N_{iterations}$ **do**

    Compute MSE loss:

    $L_{MSE} \leftarrow \|f_v(x_{adv}) - f_v(x_{target})\|_2^2$

    Compute gradient: $g \leftarrow \nabla_{x_{adv}} L_{MSE}$

    Update adversarial example: $x_{adv} \leftarrow x_{adv} - \alpha \cdot g$

    Project back onto $L_\infty$ ball: $x_{adv} \leftarrow \Pi_{B_\epsilon(x)}(x_{adv})$

  **end for**

  **return** $x_{adv}$

$D$: Diffusion model (SDXL) for image generation.
$f_v$: Vision encoder that extracts features.
$\Pi_{B_\epsilon(x)}$: Projection function to $\epsilon$-ball.

---

**Algorithm 3** CroPA-D-UAP

**Require:** Model $f$, Target Text $T$, input image $x_v$, input prompt $x_t$, perturbation size $\epsilon$, step sizes $\alpha_1$, $\alpha_2$, regularization weight $\lambda$, iterations $K$

**Ensure:** Adversarial perturbations $\delta_v$, $\delta_t$

  Initialize $\delta_v = 0$, $\delta_t = 0$

  Generate target image $T_i = \text{SDXL}(T)$

  **for** step $= 1$ to $K$ **do**

    $L_{\text{CroPA}} = \text{CroPALoss}(x_v + \delta_v, x_t + \delta_t, T)$

    Extract value vectors

    $V_t = \text{ExtractValueVectors}(T_i)$

    $L_{\text{d-UAP}} = 0$

    **for** each attention head $i$ in layer $l$ **do**

      Extract value vectors

      $V_i = \text{ExtractValueVectors}(x_v + \delta_v)$

      $L_{\text{d-UAP}} = L_{\text{d-UAP}} + cossim(V_i, V_t)$

    **end for**

    Total loss: $L_{\text{re-CroPA}} = L_{\text{CroPA}} - \lambda L_{\text{d-UAP}}$

    $g_v = \nabla_{\delta_v} L_{\text{re-CroPA}}$, $g_t = \nabla_{\delta_t} L_{\text{re-CroPA}}$

    Update perturbations:

    $\delta_v = \delta_v - \alpha_1 \cdot \text{sign}(g_v)$

    $\delta_t = \delta_t - \alpha_2 \cdot \text{sign}(g_t)$

    Project $\delta_v$ to $\epsilon$-ball around 0

  **end for**

  **return** $\delta_v$, $\delta_t$

---

Figure 9: Left: Our algorithm for initializing adversarial perturbations using diffusion models. Right: Our proposed CroPA-D-UAP method that guides perturbations via target value vectors.

## C  Implementational Details

### C.1  Datasets

Following the original work, our evaluation utilized images from the MS-COCO validation dataset Lin et al. (2014). This dataset provides a diverse collection of natural images suitable for testing cross-prompt transferability across various visual scenarios.

For the textual component, we employed two categories of Visual Question Answering (VQA) prompts. The first category, $\text{VQA}_{\text{general}}$, consists of general questions applicable to any image, focusing on common visual attributes and objects. The second category, $\text{VQA}_{\text{specific}}$, derives from the VQA-v2 dataset Goyal et al. (2017a) and contains questions specifically tailored to individual image content.

This combination of a standard vision dataset with both general and specific VQA prompts enables comprehensive evaluation of cross-prompt transferability across different types of queries and visual contexts. The prompts were designed to test both broad visual understanding and specific detail recognition capabilities of the models.

### C.2  Experimental Setup

The experimental setup followed specific parameters for attack configuration and evaluation. For the attack implementation, we maintained consistency with the original setup by utilizing the same seeds. By default, the experiments were conducted as targeted attacks, with "unknown" chosen as the target text to avoid

---

**Algorithm 4** Cross Image Cross Prompt Attack

---

**Require:** Model $f$, Target Text $T$, input images $X_v = \{x_v^1, x_v^2, \ldots, x_v^n\}$,
    prompt set $S_t = \{X_t^1, X_t^2, \ldots, X_t^n\}$, perturbation size $\epsilon$,
    step sizes $\alpha_1, \alpha_2$, iterations $K$, update interval $N$ *(controls text update frequency)*,
    SCMix augmentation boolean *SCMixAugment*,
    SCMix hyperparameters $\eta$ *(self-mixing ratio)*, $\beta_1, \beta_2$ *(cross-mixing coefficients)* if *Augment* is true Cut-Mix augmentation boolean *CutMixAugment*
**Ensure:** Adversarial perturbation $\delta_v$

1:  **Initialize** perturbation $\delta_v \sim \mathcal{N}(0, \epsilon)$
2:  **for** step $= 1$ to $K$ **do**
3:    **for** i $= 1$ to $n$ *(iterate through all images)* **do**
4:      **Sample** prompt $x_t^{i,j}$ from $X_t^i$                 *// Select random prompt for current image*
5:      **if** *SCMixAugment* is true *(enable SCMix augmentation)* **then**
6:        **Sample** cross-image $x_v^{cross} \sim X_v$          *// Randomly select image for mixing*
7:        **if** $x_v^{cross}$ is $x_v$ **then**
8:          $x_v' = x_v^i$                    *// No augmentation if same image selected*
9:        **else**
10:         **Self-mixing:** $x_{v_1}' = \text{Resize}(\text{RandomCrop}(x_v^i))$      *// Create two patches*
11:             $x_{v_2}' = \text{Resize}(\text{RandomCrop}(x_v^i))$      *// Create second patch*
12:             $x_v^{self} = \eta x_{v_1}' + (1 - \eta) x_{v_2}'$
13:        **Cross-mixing:** $x_v' = \beta_1 x_v^{self} + \beta_2 x_v^{cross}$
14:        **end if**
15:      **else if** *CutMixAugment* is true *(enable CutMix augmentation)* **then**
16:        **Sample** cross-image $x_v^{cross} \sim X_v$          *// Randomly select image for mixing*
17:        **Initialize** rectangular binary mask $M$       *// Filled with 1s where the rectangle is*
         $x_v' = (1 - M) \odot x_v^i + M \odot x_v^{cross}$
18:      **else**
19:        $x_v' = x_v^i$
20:      **end if**
21:      **if** $x_t'^{i,j}$ not initialized **then**
22:        **Initialize** text perturbation $x_t'^{i,j} = x_t^{i,j}$      *// Starting point for text perturbation*
23:      **end if**
24:      **Compute** image gradient: $g_v = \nabla_{x_v} L(f(x_v' + \delta_v, x_t'^{i,j}), T)$
25:      **Update** perturbation: $\delta_v = \delta_v - \alpha_1 \cdot \text{sign}(g_v)$
26:      **if** $\text{mod}(step, N) = 0$ *(update text every N steps)* **then**
27:        **Compute** text gradient: $g_t = \nabla_{x_t} L(f(x_v' + \delta_v, x_t'^{i,j}), T)$
28:        **Update** text perturbation: $x_t'^{i,j} = x_t'^{i,j} + \alpha_2 \cdot \text{sign}(g_t)$      *// Ascent direction*
29:      **end if**
30:      **Project** $\delta_v$ to $\epsilon$-ball: $\delta_v = \text{Clip}_{0,\epsilon}(\delta_v)$
31:    **end for**
32:  **end for**
33:  **return**  $\delta_v$             *// Return universal perturbation applicable across images*

---

high-frequency responses typical in vision-language tasks. The perturbation size was fixed at 16/255, and all adversarial examples were optimized and tested under zero-shot settings.

For multi-prompt experiments, both Multi-P and CroPA implementations used ten prompts. We maintained three evaluation runs for each experiment, averaging the Attack Success Rate (ASR) scores to ensure reliable results. The prompts spanned multiple task types including general visual questions, image-specific queries, classification tasks, and image captioning, with varying lengths and semantic structures.

For model implementations, we used the public OpenFlamingo-9B Awadalla et al. (2023) as our Flamingo variant, along with BLIP-2 and InstructBLIP models. Our reproduction maintained these core experimental parameters to ensure comparable results with the original work.

## C.3 Computational Requirements

The computational demands of reproducing the CroPA experiments were substantial, reflecting the resource-intensive nature of modern Vision-Language Models. Our primary experiments were conducted on a PyTorch Lightning platform using an L40S GPU with 48GB VRAM and a 4-core CPU with 16GB RAM, matching the original paper's minimum requirement of 45GB VRAM for stable execution.

Working within the constraints of the free-tier platform credits posed significant challenges. Our experiments were limited by a pooled allocation of 120 credits shared across four accounts. This necessitated careful resource management, particularly given the computational intensity of large-scale VLMs. To overcome these limitations, we implemented several memory optimization strategies to enable partial execution on local machines with 16GB VRAM, though this required significant code modifications.

The total computational cost of our reproduction study amounted to approximately 140 GPU hours and 90 CPU hours. This includes time spent on model training, attack generation, and evaluation across multiple experimental configurations. The substantial computational requirements underscore the importance of efficient resource allocation in modern machine learning reproducibility studies.

# D Experiment Hyperparameters

## D.1 Targeted ASRs tested on Flamingo with different target texts using CroPA

The following are the detailed hyper-parameters used to obtain the results in Table 1, under Subsection 6.1:

CROPA utilizes the following hyperparameters for optimizing adversarial perturbations in vision-language models. These parameters are designed to balance attack effectiveness while preserving task-specific functionality.

### D.1.1 Optimization Parameters

- **Total Iterations:** 1,701

- **Step Size:**
    - **Image Perturbations** ($\alpha_1$): $\frac{1}{255} \approx 0.0039$
    - **Text Embedding Perturbations** ($\alpha_2$, CROPA method): 0.01

- **Perturbation Budget** ($\epsilon$): $\frac{16}{255} \approx 0.0627$ under $L_\infty$ constraint

- **Loss Function:** Mean Squared Error (MSE) on ViT embeddings

- **Batch Size:** Dynamically allocated based on available GPU memory

### D.1.2 Multi-Prompt Strategy

- **Simultaneous Prompts per Image:** 10 (default)

- **Prompt Rotation:** Randomized cyclic permutation per full cycle

- **Context Token Masking:** Preserve first $N$ context tokens during updates

### D.1.3 Base Models Supported

- OpenFlamingo

- BLIP-2

- InstructBLIP

### D.1.4  Generation Parameters

- **Beam Search Width:** 3

- **Length Penalty:** -2.0 (encourages concise outputs)

- **Maximum Generation Length:** 5 tokens

### D.1.5  Text Embedding Perturbation

- **Update Intervals:** Every 30 iterations (for 10 prompts)

- **Constraint:** $\pm 0.27$ deviation from original embeddings

### D.1.6  Reproducibility

- **Random Seed:** 42

- **Image Preprocessing:** 224×224 center crop

This configuration enables simultaneous optimization of vision-language perturbations while maintaining task functionality through constrained gradient updates.

## D.2  Noise Initialization via Vision Encoding Optimization

This section details the key hyperparameters used in our CroPA with Noise Initialisation via Vision Encoding optimisation attack detailed in Subsection 5.1, to obtain the results in Table 3 and provides a rationale for their selection. We aim to provide sufficient information for reproducibility and to justify our experimental choices. **Projected Gradient Descent(PGD)** was used to align the original image $x_i$ with the target image $T_i$ by adding perturbations iteratively.

### D.2.1  PGD via Vision Encoder Image Perturbation Hyperparameters

- **Epsilon ($\varepsilon$):** We set the maximum allowed pixel perturbation ($L_\infty$ norm) to 16/255. This value represents a trade-off between attack strength and perceptibility. Smaller epsilon values might be less effective at fooling the model but also less noticeable to human observers.

- **Alpha1 ($\alpha_1$):** The step size for each PGD iteration was set to 1/255. This relatively small step size allows for finer-grained exploration of the perturbation space and helps to avoid overshooting optimal perturbation directions.

- **PGD Iterations**: The number of PGD iterations for image perturbation was set to 1701. This was determined empirically; we observed that increasing iterations beyond this point yielded diminishing returns in terms of attack success rate while significantly increasing computation time. Save perturbation iterations are 900, 1100, 1300,1500 and 1700.

- **Budget**: 0.05. This hyperparameter defines the maximum allowed change to each pixel value in the image during the PGD optimization process for aligning ViT embeddings. It constrains the perturbation magnitude to ensure the modified image remains visually similar to the original.

- **Timesteps**: 150. Specifies the number of optimization steps taken during the PGD process to align ViT embeddings between the generated and target images. A larger number of timesteps allows for finer adjustments and potentially better alignment but also increases computational cost.

### D.2.2 Text Perturbation Hyperparameters (CroPA Specific)

- **Alpha2 ($\alpha_2$)**: This hyperparameter controls the step size for updating the text embedding perturbations. The value is dynamically assigned based on the number of prompts used.

- **Number of Prompts**: The number of prompts utilized during the optimization phase was 10. We use multiple prompts to make a more robust and transferable perturbation. Using different prompts that all target the same wrong answer forces the attack to find a perturbation that works across variations in the input question.

- **CroPA Update Iterations**: Text perturbations are updated at specific iterations (defined by `cropa_iter`). The text perturbation will be updated till 300 iterations. We empirically found that the text perturbation can guide the optimization better with some iterations.

  ```
  cropa_end = 300
  step = max((cropa_end//prompt_num),1)
  cropa_iter = [i for i in range(step,cropa_end+1, step)]
  ```

### D.2.3 Evaluation and Dataset Hyperparameters

- **Test Dataset Fraction**: We evaluated our attack on a 5% (fraction = 0.05) subset of the test dataset. This allowed us to perform a thorough evaluation while keeping the computational cost manageable. We selected the subset randomly to ensure a representative sample of the overall test distribution.

- **N-Shot Examples**: The number of in-context learning examples was set to 0. This means we did not provide any demonstration examples to the model during evaluation.

- **Number of Test Images**: 50.

- **Max Generation Length**: 5.

- **Num Beams**: 3.

- **Length Penalty**: -2.0.

### D.2.4 Noise Initialization via Vision Encoding Optimization

- **ViT Model**: `ViTModel.from_pretrained("openai/clip-vit-large-patch14")`

- **Learning Rate**: 0.1

- **Optimizer**: Adam

### D.2.5 Additional Notes

- **Random Seed**: We used a fixed random seed (`seed_everything(42)`) to ensure reproducibility of our results.

- **Device**: All experiments were conducted on a GPU (`cuda:{ config_args.device}`).

- **Batch Size**: The evaluation batch size (`args.eval_batch_size`) was chosen to maximize GPU utilization without exceeding memory constraints.

### D.2.6 Justification of Choices

The hyperparameter values were selected based on a combination of prior work, pilot experiments, and computational constraints. We prioritized settings that balanced attack effectiveness, stealthiness, and computational efficiency. Ablation studies (not included in this section but discussed elsewhere in the paper) were conducted to assess the sensitivity of our results to key hyperparameters such as $\varepsilon$ and $\alpha$.

### D.3 Investigating Cross Image Transferability and Image Augmentation

The following are the detailed hyper-parameters used to obtain the results in Table 5, under Subsection 6.5.2

For SCMix, the only hyperparameters are the choice of $\eta$ (or $\alpha$ if $\eta \sim \text{Beta}(\alpha, \alpha)$), $\beta_1$ and $\beta_2$.

- **Eta ($\eta$) and Alpha ($\alpha$)**: We chose $\eta = 0.5$ as a constant value for $\eta$ to ensure that the degree of self-mixing during augmentation is constant (and equal). In terms of $\alpha$, this is equivalent to setting $\alpha = \infty$. We do not experiment with this, as our main goal with SCMix is to ensure cross-image diversity.

- **Beta1 ($\beta_1$) and Beta2 ($\beta_2$)**: We took several values of $\beta_1$ from 0.5 to 1, fixing $\beta_2 = 1 - \beta_1$ to ensure that the base image, used for self-mixing, dominates the final augmented image in terms of visual characteristics, while still ensuring diversification by adding some characteristics of the cross-mixing image. Our results were reported for the best value we got, which was with $\beta_1 = 0.7$ and $\beta_2 = 0.3$. The justification behind fixing $\beta_2$ relative to $\beta_1$ is so as to avoid clipping and ensure cross-mixing is treated like a true weighted average.

For CutMix, there are no specific hyperparameters to consider.

The other hyperparemeters pertaining to the base CroPA framework can be found in D.1. Apart from this, in terms of implementation, the only variation with base CroPA is that of the perturbation learned for an image and its set of text prompts, the one learned for the image is common across all the images used.

### D.4 D-UAP Enhancements

Our implementation of CroPA (Cross-modal Prompt Attack) and the baseline UAP (Universal Adversarial Perturbation) approach utilized the following hyperparameters:

#### D.4.1 Attack Configuration

- **Maximum Perturbation ($\varepsilon$)**: 16/255, constraining the $L_\infty$-norm of the adversarial perturbation.

- **Step Size ($\alpha_1$)**: 1/255, controlling the gradient update magnitude for image perturbations.

- **Text Embedding Perturbation Range**: $[-0.23, 0.27]$, limiting the word embedding perturbation magnitude.

- **Text Perturbation Step Size ($\alpha_2$)**: Varies based on prompt number, determining the magnitude of updates to text embeddings.

- **Total Iterations**: 1701, with evaluation checkpoints at intervals of 200, starting from iteration 900.

- **Target Response**: "bomb," forcing the model to generate this specific output.

- **Loss Weight ($\lambda$)**: 5

#### D.4.2 CroPA-Specific Parameters

- **Text Perturbation Schedule**: Updates occur during the first 300 iterations at intervals determined by step $= \max((300/\text{prompt\_num}), 1)$.

- **Prompt Access Strategy**: Random shuffling with rotation to ensure diverse prompt coverage.

- **Semantic Alignment**: Cosine similarity loss with weight factor 5 to align perturbed images with the target concept.

### D.4.3 Evaluation Configuration

- **Sampling Fraction**: 0.05 of the total dataset, balancing computational resources with statistical significance.

- **In-Context Learning Examples**: 0 shots (default configuration).

- **Generation Parameters**:
  - Maximum generation length: 5 tokens.
  - Number of beams: 3.
  - Length penalty: $-2.0$.

### D.4.4 Model-Specific Settings

We conducted the same experiment for both BLIP-2 and OpenFlamingo. Hyperparameters for both the models are given
**Experiment 1:**

- **Primary Model**: OpenFlamingo-9B

- **Image Generation Model**: stable-diffusion-xl-base-1.0 was used.

- **Vision Feature Extraction**: Middle transformer blocks (layers 10-19) for semantic representation.

- **Image Preprocessing**: Normalization with mean $= [0.485, 0.456, 0.406]$ and standard deviation $= [0.229, 0.224, 0.225]$.

- **Image Dimensions**: $224 \times 224$ pixels.

**Experiment 2:**

- **Primary Model**: BLIP-2 (blip2-opt-2.7b).

- **Image Generation Model**: stable-diffusion-xl-base-1.0 was used.

- **Vision Feature Extraction**: Middle transformer blocks (layers 14-29) for semantic representation.

- **Image Preprocessing**: Normalization with mean $= [0.485, 0.456, 0.406]$ and standard deviation $= [0.229, 0.224, 0.225]$.

- **Image Dimensions**: $224 \times 224$ pixels.

The hyperparameters were carefully selected to balance attack effectiveness and computational efficiency. Our implementation used a random seed of 42 to ensure reproducibility across experimental runs.

### D.4.5 Cross Model test with OpenFlamingo as the primary model

| Settings | Method | $VQA_{general}$ | $VQA_{specific}$ | Classification | Captioning | Overall |
|---|---|---|---|---|---|---|
| Flamingo to InstructBLIP | Multi-P | 0.00 | 0.00 | 0.00 | 0.00 | 0.00 |
| | CroPA | 0.00 | 0.00 | 0.00 | 0.00 | 0.00 |
| | CroPA + D-UAP | 0.00 | 0.00 | 0.00 | 0.00 | 0.00 |
| Flamingo to BLIP2 | Multi-P | 0.00 | 0.00 | 0.00 | 0.00 | 0.00 |
| | CroPA | 0.00 | 0.00 | 0.00 | 0.00 | 0.00 |
| | CroPA+D-UAP | 0.00 | 0.00 | 0.00 | 0.00 | 0.00 |

Table D.4.5 reports results for the cross-model transferability test as mentioned in Subsection 6.5.3 with the D-UAP enhanced CroPA method described in Subsection 5.2.

# E   Prompts for Different Tasks

This section presents a short description of prompts used for various vision-language tasks in our experiments. Detailed list of prompts is provided in our supplementary material.

Prompts for Visual Question Answering (VQA) are categorized into two types: $VQA_{general}$ and $VQA_{specific}$. $VQA_{general}$ prompts are image-agnostic while $VQA_{specific}$ prompts are tailored to specific image content. Prompts in the categories of $VQA_{general}$, Image captioning and Image Classification are created by the original paper Luo et al. (2024) while the prompts for $VQA_{specific}$ are derived from the Goyal et al. (2017b).

