# OpenReview forum: "Revisiting CroPA: A Reproducibility Study and Enhancements for Cross-Prompt Adversarial Transferability in Vision-Language Models"
_TMLR — Accepted by TMLR_

### Review · Reviewer_cw34 · 2025-03-16

**Summary Of Contributions:**

This submission investigates the transferability of adversarial attacks across different models within the Cross-Prompt Attack (CroPA) framework. The work first replicates and verifies key aspects of the original CroPA study, confirming most of its reported findings. Beyond replication, the study makes two notable contributions to enhance cross-model transferability:
- Instead of relying on conventional random noise, the authors propose leveraging text-to-image diffusion models to generate structured initial perturbations.
- The study integrates input diversity and Universal Adversarial Perturbation (UAP) techniques, which have been previously effective in image-based adversarial attacks.

**Audience:**

Yes

**Broader Impact Concerns:**

This work highlights critical adversarial vulnerabilities in VLMs, which can aid in developing stronger defenses but also risk misuse for generating more effective attacks. The findings emphasize the need for improved robustness in VLMs.

**Claims And Evidence:**

Yes

**Requested Changes:**

As mentioned earlier, the suggested revisions are listed below in order of importance. These modifications are crucial for the acceptance of this work. I believe that upon completing these revisions, this work will be worthy of acceptance.

1. Enhance the discussion on the potential trade-offs between cross-model, cross-prompt, and cross-image transferability.

2. Expand cross-model transferability experiments by dedicating a section to the threat model and providing more comprehensive evaluations beyond Table 6.

3. Provide further analysis on why the noise initialization strategy plays a crucial role in CroPA and whether this is due to the bi-level optimization framework. Discuss if alternative initialization strategies could further improve performance.

4. Justify the use of LD-DUAP, addressing concerns about overfitting to the surrogate model.

5. Fix formatting issues in Table 6 for better readability.

**Strengths And Weaknesses:**

### **Strengths**

- **Clarity and Readability**: The paper is well-written and easy to follow, making it accessible to a broad audience in adversarial machine learning.
- **Practical and Important Research Problem**: The study addresses a highly relevant issue—**adversarial vulnerabilities in Vision-Language Models (VLMs)**—which has significant implications for both security and robustness in multimodal AI systems.
- **Insightful Identification of Cross-Model Transferability**: The authors astutely recognize **cross-model transferability** as a key challenge in adversarial attacks against VLMs. While CroPA briefly mentioned this concept in its original paper, this work delves deeper into it as a primary research focus. However, the paper would benefit from a more thorough conceptual discussion of this issue. E.g., is there a **trade-off** between **cross-model transferability, cross-prompt transferability, and cross-image transferability**?
- **Significance of the Proposed Noise Initialization Strategy**: Although the idea of replacing random noise initialization is not entirely novel, its impact in the context of CroPA) is significant. Traditionally, in adversarial attacks on computer vision models, initialization strategies do not substantially affect results. However, this study demonstrates that in CroPA, **the choice of initialization plays a crucial role**. This insight could lead to further investigations into:
  - Whether the importance of initialization stems from the bi-level optimization framework of CroPA, which makes learning more difficult.
  - In standard CV adversarial attacks, adversarial perturbations naturally adopt the target texture during training. If this does not happen in CroPA, it suggests that the **optimization process itself may have room for significant improvement**, beyond just modifying the initialization strategy.

### **Weaknesses**

- **Limited Experimental Validation of Cross-Model Transferability**: Since this work **centers around cross-model transferability**, it should provide more extensive experiments to support its claims. However, the only relevant experimental results are found in Table 6.
  - To align with best practices in adversarial ML research, the authors should dedicate an entire section*to defining their **threat model** and clearly detailing their cross-model transferability experiments.
  - Additional evaluations on a broader range of models and architectures would significantly strengthen the empirical contributions.
- **Questionable Motivation for the LD-DUAP Strategy**: The second proposed strategy—LD-DUAP—raises concerns:
  - The authors acknowledge that UAP technology primarily enhances cross-image transferability, which can lead to overfitting to the surrogate model (white-box model).
  - This overfitting could actually hurt cross-model transferability, contradicting the study's primary objective.
  - The paper should explicitly justify why LD-DUAP is beneficial in this context and provide ablation studies to verify its effectiveness in improving cross-model transferability rather than harming it.

---

> ### Author Response · Authors · 2025-05-01
> **Response to Reviewer cw34 (Part 1/4)**
>
> **W1 & RC2 Limited Experimental Validation of Cross-Model Transferability**
> > Expand cross-model transferability experiments by dedicating a section to the threat model and providing more comprehensive evaluations beyond Table 6; Additional evaluations on a broader range of models and architectures would significantly strengthen the empirical contributions.
>
> Thank you for this insightful and constructive feedback. In response to your suggestion, we have significantly expanded our empirical evaluation by including additional vision-language models: **LLaVA and InstructBLIP** into our analysis, complementing the existing evaluations on **BLIP-2 and Flamingo**.** We conducted a full suite of cross-model experiments across all four models, evaluating transfer performance. The updated results are presented in **Table 8**, which now reflects a broader and more rigorous validation of our proposed methods.
>
> Additionally we would like to put forward that while cross-model transferability was a critical cornerstone of our work, it does not fully represent the all the central ideas put forth in it. Additional core contributions in our work represent investigations into **cross-image transferability** using data augmentations as well, proposing method incorporations to generalize the perturbations across images in addition to the original motivation of CroPA to generalize across prompts.
>
> Extensive analysis of these methods and their motivations, along with the limitations of CroPA that fueled the same, can be found in Section 5.3, which introduces SCMix and CutMix as image augmentation techniques that could lead to these enhancements derived from ideas in UAP literature. We also specify the Threat Model for this setup and Evaluation Rationale comprehensively in sub-subsection 5.4.2. Empirical evidence presented in sub-subsection 6.5.2, supports our hypotheses regarding the methods and further discussion in 7.3.1 evaluate the strengths and limitations of our experimental methods, clearly underscoring corresponding mitigated limitations in CroPA.
>
> Apart from cross-image transferability, another core objective focused on recognizing CroPA’s initialization dependence and attempting to mitigate the challenges caused by it, providing an in-detail analysis as well.
>
> We also deeply appreciate your recommendation to clearly define our threat model and experimental setup. Following this suggestion, we have added an extensive new section (**Section 5.4)** dedicated to detailing our threat model, adversarial goals, model access levels and evaluation setup for cross-model as well as cross image transferability. For Cross-Model scenarios, we differentiate between Intra-family and Cross-Architecture Transfer noting that since BLIP2 and InstructBLIP have similar vision encoders, transferability rates between these models are higher. We also add the Evaluation Rationale (SubSection 5.4.3) for our experiments, since we present our transferability evaluations in an untargeted setting, aligning with standard practices in the adversarial transferability literature.  We believe these additions significantly strengthen the empirical and conceptual clarity of the paper.

---

> ### Author Response · Authors · 2025-05-01
> **Response to Reviewer cw34 (Part 2/4)**
>
> **W2 and RC4 Motivation for the DUAP Strategy**
>
> We thank the reviewer for their thoughtful comments regarding the motivation and effectiveness of our DUAP strategy. We acknowledge that due to different architectures of LVLMs, overfitting to the surrogate white-box model can be harmful for generalization, however But we will indeed like to clarify here that **overfitting to the surrogate model’s architecture and coupling with the low-level features of the image is an inherent characteristic of CroPA** rather than the UAP methods. Our approach with using UAPs was derived from the motivation against CroPA’s overfitting limitations, and to make it’s perturbations transferable across **both** paradigms, cross-image and cross-model, in order to create an attack to be truly transferable across all three paradigms.
>
> For cross-image enhancements we propose incorporations to generalize the perturbations across images in addition to the original motivation of CroPA to generalize across prompts, extensive analysis of these methods and their motivations, along with the limitations of CroPA that fueled the same, can be found in Section 5.3, which introduces SCMix and CutMix as image augmentation techniques that could lead to these enhancements derived from ideas in UAP literature (As discussed previously, the details are now extensively updated in 5.3, 5.4.2, 6.5.2 and 7.3.1).
>
> For our cross-model enhancement, our approach is strategically motivated by the architectural commonality across modern LVLMs: they typically employ Vision Transformers (ViTs)  as their image encoders to convert images into visual token [3][4]. The DUAP strategy specifically targets the encoded visual tokens produced by these ViTs, which exhibits global effectiveness across different LVLMs for several reasons:
>
> 1. Vision-language models share similar tokenization strategies despite architectural differences in their language components. The initial image processing stage has significant architectural consistency across models.
> 2. Recent research on visual token pruning in LVLMs demonstrates that visual tokens from the ViT encoder are processed in a relatively consistent manner across different model architectures. [2]. This consistency extends to adversarial vulnerability.
> 3. Studies on adversarial transferability across vision-language models have shown that attacks targeting the visual encoder component can transfer effectively between different model architecture. [1][3][4]
>
> Our targeting of value vectors is particularly strategic because these vectors hold the actual information content within each image patch, thus  perturbing these vectors directly impacts the information content being processed. By disrupting these value vectors to align with those of a **target image,** a visual proxy for the target text, we are not overfitting to model-specific gradients, but instead attacking a **shared representational space.** By targetting our attack to only the vision encoder, we nullify the effect of the difference in architectures of the different LVLMs. This is also explained in detail in the now updated Section 5.2 and Subsection 7.3.2.
>
> In our experiments, we investigate the cross-model transferability, taking BLIP2, InstructBLIP and Flamingo as surrogate models and investigating the transferability of their attacks against each other as well as LLaVA (Subsection 5.4.1 and 6.5.4). We demonstrate that our DUAP strategy, significantly improves cross-model transferability (Table 8). However,  the method’s effectiveness is constrained by its dependency on the model’s having similar vision encoders. Thus, models like BLIP2 and InstructBLIP which share the same vision encoder show greater transferability to each other than to decoder-only models such as LLaVA and Flamingo which have similar but non-identical vision encoders.
>
> [1] Yubo Wang, Chaohu Liu, Yanqiu Qu, Haoyu Cao, Deqiang Jiang, and Linli Xu. Break the visual perception: Adversarial attacks targeting encoded visual tokens of large vision-language models, 2024
>
> [2] Kai Huang, Hao Zou, Ye Xi, BoChen Wang, Zhen Xie, and Liang Yu. 2024. IVTP: Instruction-Guided Visual Token Pruning for Large Vision-Language Models. In Computer Vision – ECCV 2024: 18th European Conference, Milan, Italy, September 29–October 4, 2024, Proceedings, Part XVII
>
> [3] D. Lu et al., "Set-level Guidance Attack: Boosting Adversarial Transferability of Vision-Language Pre-training Models," in 2023 IEEE/CVF International Conference on Computer Vision (ICCV), Paris, France, 202
>
> [4] Christian Schlarmann, Naman Deep Singh, Francesco Croce, and Matthias Hein. 2024. Robust CLIP: unsupervised adversarial fine-tuning of vision embeddings for robust large vision-language models

---

> ### Author Response · Authors · 2025-05-01
> **Response to Reviewer cw34 (Part 3/4)**
>
> **RC1: Enhance the discussion on the potential trade-offs between cross-model, cross-prompt, and cross-image transferability.**
>
> Thank you for this helpful suggestion. We have updated our paper and added a dedicated section 7.4 to explicitly discuss the interactions and potential tensions between **cross-model**, **cross-prompt**, and **cross-image** transferability. This section highlights key insights from our empirical analysis and outlines how different strategies and ablations in our framework contribute to balancing or improving across these axes.
>
> In particular, we show that the original CroPA method prioritizes cross-prompt effectiveness by anchoring perturbations to model-specific text embeddings, but this approach inherently limits cross-model generalization and fails entirely for cross-image transferability by tightly coupling perturbations to image-specific features. We develop strategies to directly address these limitations.
>
> Our **DUAP strategy** improves **cross-model** transferability while maintaining **cross-prompt** transferability by targeting shared components within vision encoders rather than model-specific features generating more robust and generalizable perturbations. Meanwhile, our **image augmentation strategy**  (SCMix and CutMix) enhances **cross-image** and **cross-prompt** generalization by introducing spatial diversity during training. Finally, we find that by anchoring perturbations to semantically meaningful initializations, our **perturbation initialization strategy** helps achieve a **higher overall attack success rate (ASR)** without compromising transferability in any of the three dimensions. This suggests that the optimization landscape's starting point significantly influences the achievable balance between different transferability types.

---

> ### Author Response · Authors · 2025-05-01
> **Response to Reviewer cw34 (Part 4/4)**
>
> **RC3 Noise initialization strategy analysis**
> > Provide further analysis on why the noise initialization strategy plays a crucial role in CroPA and whether this is due to the bi-level optimization framework. Discuss if alternative initialization strategies could further improve performance.
>
> As you rightly pointed out, we investigated if CroPA’s bi-level optimization is susceptible to initialization changes, as is usually the case with bi-level optimizations on complex task-surfaces, sub-subsection 6.5.1 is now updated  theoretically verify this by extensive literature review and empirical evidence in sub-subsection 6.5.1 under the paragraph titled “Theoretically Analysis of Initialization in Bi-Level  Optimization”, and conclusive evidence was provided by our results under 6.5.1 with Table 4 and Figure 7,  and the results of our improved initialization depicted in Table 3, showing consistent gains over the CroPA and the baselines. The main identified points were :
>
> 1. CroPA's min-max formulation exhibits high sensitivity to initialization conditions, requiring carefully designed starting points for stable convergence across the saddle-point optimization landscape [1].
> 2. The asymmetric update schedule between image and prompt perturbations creates local cycling behaviors, making convergence heavily dependent on initial proximity to adversarially robust regions.
> 3. Semantic initialization provides superior cross-model transferability compared to random initialization, particularly effective in complex bi-level optimization contexts where pathways are less direct [2].
>
> We have strengthened the justification for our claim by expanding Section 5.1 and empirically validating the impact of initialization on convergence and ASR performance. Specifically:
>
> - We now explicitly articulate a core limitation of CroPA and similar gradient-based methods: their reliance on random noise for initialization. This approach, while widely adopted, lacks semantic guidance and fails to align with the structured multimodal input space of VLMs.
> - Although CroPA refines this noise iteratively, our investigation shows that an uninformed starting point introduces instability and results in slower or suboptimal convergence—particularly critical in VLMs, where semantic alignment is essential for effective attacks.
> - To substantiate this, we have conducted an in-depth empirical analysis in Section 6.5.1. The updated Figure 7 and Table 3 clearly demonstrate CroPA’s sensitivity to initialization. The results confirm that our proposed initialization strategy not only enhances convergence speed but also yields consistently higher ASR.
>
> The initialization also is motivated by limitations now updated in Section 5.1 :
>
> CroPA, like many gradient-based adversarial attack methods, uses random noise initialization for perturbations, which is agnostic to the semantic structure of the target prompt. This uninformed starting point leads to instability and slower convergence, especially problematic in vision-language models where semantic alignment with prompts is crucial.  Our comprehensive evaluation (Section 6.5.1, Figure 7, Table 3) empirically demonstrates that CroPA’s effectiveness is highly sensitive to initialization, with random noise failing to guide optimization toward semantically meaningful directions from the outset. As a result, attacks initialized randomly require more optimization steps to achieve alignment with the target prompt’s semantics, often resulting in suboptimal attack success rates and reduced cross-prompt transferability.  Our initialization strategy ensures that adversarial optimization begins in a semantically meaningful neighborhood, directly addressing CroPA’s instability and inefficiency due to random initialization.
>
> We believe these additions offer a clearer understanding of CroPA’s limitations and strongly support our position regarding the role of initialization in adversarial optimization for VLMs.
>
> **RC5: Formatting issues in Table 6**
>
> Thank you for pointing this out, we have re-formatted all the tables throughout the paper, to ensure consistency and readability.
>
> [1] Zhang, J, Xiao, P, Sun, R & Luo, ZQ 2020, 'A single-loop smoothed gradient descent-ascent algorithm for nonconvex-concave min-max problems', Advances in Neural Information Processing Systems
>
> [2] Dongxian Wu, Shu-Tao Xia, and Yisen Wang. 2020. Adversarial weight perturbation helps robust generalization. In Proceedings of the 34th International Conference on Neural Information Processing Systems (NIPS '20)
>
>
> We have implemented the suggested changes to the bets of our knowledge and ability. Kindly let us know if they fulfill your concerns appropriately.

---

### Review · Reviewer_3isH · 2025-04-10

**Summary Of Contributions:**

This paper presents a comprehensive reproducibility study and enhancements to the Cross-Prompt Attack (CroPA) for Vision-Language Models (VLMs). The authors first reproduced the results of the original CroPA paper, validating its cross-prompt transferability and overall performance. Building on this foundation, they propose several key improvements:

 - Noise Initialization via Diffusion Models: A novel initialization strategy that leverages diffusion models to generate semantically informed perturbations, significantly boosting attack success rates.

 - Guided Perturbations via Vision Encoder: Integration of targeted value vector manipulation in the vision encoder’s attention mechanism to enhance adversarial effectiveness.

 - Cross-Image Transferability with ScMix: Exploration of Universal Adversarial Perturbations (UAPs) using CroPA’s framework, augmented by ScMix data augmentation to improve generalization across images.

The enhancements demonstrate consistent improvements in adversarial performance while highlighting challenges in cross-model and cross-image transferability. The study underscores the vulnerabilities of VLMs to adversarial attacks and provides insights for future robustness research.

**Audience:**

Yes

**Claims And Evidence:**

Yes

**Requested Changes:**

1. Validate LD-UAP performance claims (Clarify the 11.6% ASR improvement claim vs. Table 5's 0.2% difference)
2. Explain limited VQA improvements (Add analysis of why CroPA shows marginal gains in VQA tasks)
3. Expand ScMix analysis
 - Test more hyperparameters (e.g., η∼Beta(0.5,0.5))
 - Compare to simpler methods like CutMix)
4. Certain typo fix.

**Strengths And Weaknesses:**

Strengths
1. Rigorous Reproducibility Study
   - Comprehensive validation of CroPA across multiple VLMs (Flamingo, BLIP-2, InstructBLIP) with consistent results.
   - Clear documentation of hyperparameters and experimental settings, enhancing reproducibility.

2. Clear and Detailed Writing
   - Well-structured methodology with sufficient technical depth (e.g., diffusion-based initialization in Section 4.1).
   - Transparent reporting of limitations (e.g., computational constraints in Section 6.1).

Weaknesses
1. Limited Impact of Image Augmentation Without Further Exploration of the Reasons
   - ScMix yields marginal improvements in cross-image transferability (Table 4).

2. VQA Performance Discrepancy Without Further Exploration of the Reasons
   - CroPA + LD-UAP shows limited gains over baselines in VQA tasks (Table 5)

3. Unclear LD-UAP Improvements
   - Claims of "11.6% ASR increase" (Section 5.5.3) are unsupported by Table 5 (95.4% → 95.6%).

4. Minor Issues
   - Typographical errors (e.g., "LD-DUAP" → "LD-UAP").

---

> ### Author Response · Authors · 2025-05-01
> **Response to Reviewer 3isH (Part 1/2)**
>
> **W1:** **Limited Impact of Image Augmentation**
> > Limited Impact of Image Augmentation Without Further Exploration of the Reasons
>
> Thank you for bringing the limited gains from SCMix to our notice. In the previous version of our submission, we evaluated cross-image transferability using *targeted* attack metrics. However, we have since updated our evaluation to report *untargeted* attack success, which is more aligned with standard practices in the cross-image transfer literature. We have also incorporated another augmentation method, CutMix, as per your recommendation. Under this more appropriate evaluation, both SCMix and CutMix lead to significantly improved cross-image transferability, as shown in the updated Table 5. We believe the previously limited performance in the targeted setting arises from the inherent nature of image augmentations like SCMix and CutMix. These methods blend features from two unrelated images, which introduces spatial and semantic noise that can disrupt the alignment needed for a targeted adversarial perturbation to succeed on a different image. Untargeted attacks are more tolerant to this kind of noise, as they only require deviation from the original prediction rather than convergence to a specific alternative. This makes SCMix and CutMix more effective in the untargeted setting, where their ability to improve generalization of perturbations across image space becomes more apparent. Additional details on our evaluation rationale and subsequent discussion of the results can be found in Subsection 5.4.3 and Subsection 7.3.1.
>
> **W2 & W3 & RC1: Validate LD-UAP performance claims and justify limited VQA improvements**
> > VQA Performance Discrepancy Without Further Exploration of the Reasons; Unclear LD-UAP Improvements
>
> Thank you for pointing this out. In the original version of the paper, we mistakenly used different target texts while reporting the VQA results in Table 5, specifically, "unknown" for CroPA and Multi-P, and "bomb" for CroPA with DUAP. This was an unfortunate and rare oversight in our evaluation process. Upon discovering the inconsistency, we thoroughly reviewed and re-ran all relevant experiments to ensure consistency across all comparisons. We also performed additional evaluations on multiple models to confirm the reliability of our updated results as is evident in Table 6,7. With the corrected and expanded analysis, we observe that CroPA combined with LD-UAP consistently outperforms standard CroPA in each setting including VQA, thereby supporting our original claims more robustly.

---

> ### Author Response · Authors · 2025-05-01
> **Response to Reviewer 3isH (Part 2/2)**
>
> **RC3: Expand SCMix analysis**
> >Expand ScMix analysis; Test more hyperparameters (e.g., η∼Beta(0.5,0.5)); Compare to simpler methods like CutMix)
>
> Thank you for the constructive suggestions. Initially, we only choose SCMix as it had empirical results to support it as a suitable augmentation method for pre-trained VLPs. However, based on your feedback, we expanded our analysis by adding a direct comparison with CutMix, which has now been included in our additional investigative methods for cross-image transferability in 5.3.2.  We present the empirical obtained by included CutMix in the comparison in Table 5 under 6.5.2, and recognize that the inclusion of simpler methods like CutMix was also valuable, since it demonstrated better performance over the base method, with the following results :
>
> - We discovered that CroPA's adversarial perturbations are tightly coupled to specific image features, limiting cross-image transferability despite the original authors' claims.
> - Our implementation of both SCMix and CutMix augmentation techniques as additions to CroPA successfully improved cross-image transferability while preserving semantic information.
> - SCMix showed the strongest performance with approximately 27% ASR improvement overall, performing particularly well in VQA and captioning tasks.
> - CutMix provided a more modest but still significant 12% improvement over base CroPA in cross-image scenarios
>
> We present a detailed discussion on both of their  respective strengths and improvements shown over CroPA in 7.3.1 where we focus on :
>
> - **Methodological differences**: SCMix employs a two-stage mixing approach (self-mixing followed by cross-mixing) with continuous blending, while CutMix uses a simpler, discrete region-based replacement strategy.
> - **Mechanism of improvement**: SCMix effectiveness stems from its introduction of spatial and scale invariance through random cropping and resizing, making perturbations less sensitive to exact positioning. CutMix's strength lies in its preservation of distinct, non-overlapping image regions.
>
> We are grateful for you pointing out this valuable inclusion. Implementing  CutMix captured a broader scope it represents a fundamentally **complementary augmentation philosophy compared to SCMix** - using hard regional replacement rather than soft blending. This methodological contrast allows us to directly test whether discrete or continuous mixing strategies are more effective for transferable adversarial perturbations. Since other simpler augmentations (eg.: MixUp) follow a similar concept to CutMix,  they will likely follow a similar trend. While we do  acknowledge that comparing with additional methods would provide a more comprehensive evaluation c**omputational constraints** led us to prioritise capturing the widest scope with our analysis with these two methods showing key comparison that offers clear insights into how different mixing approaches affect cross-image transferability.
>
> As per your suggestion **regarding hyper-parameter testing**, we ran SCMix for sweeping different values of the hyper-parameter beta_1 (and beta_2), and have reported the optimal results obtained by sweeping their values. However, we did not experiment with our choice of alpha (and eta), keeping it fixed, and the rationale behind these choices is presented in the Appendix Subsection D.3, where we clarify that since our main goal with SCMix is to ensure cross-image diversity, and eta being a **self-mixing parameter**, we take the recommended value to ensure that the degree of self-mixing during augmentation is constant (and equal). We have also updated these results and configurations in Table 5 and included a more detailed methodology on Image Augmentation methods in Subsection 5.3.  We have also fixed the naming convention for the SCMix method across the paper, in order to avoid potential mismatches and confusion.
>
> **W4: Typographical errors (e.g., "LD-DUAP" → "D-UAP").**
> >Certain typo fix.
>
> Thank you for pointing this out, we have corrected the typo. The the ‘L’ was supposed to represent the custom loss we introduced into the CroPA method, which need not be included in the method's name.
>
>
> We have noted and implemented the requested changes to the best of our knowledge and ability, and we sincerely hope these changes address the concerns over the paper

---

### Review · Reviewer_PcXE · 2025-04-14

**Summary Of Contributions:**

This paper presents a comprehensive study of the previously proposed CroPA method to ensure reproducibility. Additionally, it introduces several novel techniques to further enhance the performance of CroPA.

**Audience:**

Yes

**Claims And Evidence:**

Yes

**Requested Changes:**

see weakness

**Strengths And Weaknesses:**

Strengths:

1. The ideas behind the new techniques such as the novel initialization strategy and the use of target value vectors to guide perturbations are well-motivated.
2. The proposed techniques demonstrate promising results, indicating their effectiveness in enhancing the original CroPA method.

Weakness:

1.  While reproducing previous work is important, it should not be positioned as a key contribution of the paper. Instead, the authors should focus on clearly identifying the limitations of the original CroPA method and emphasize how these shortcomings motivate the development of their proposed improvements.
2. The experimental evaluation of the proposed methods appears insufficient. For example, in Table 3, only three target prompts are used for evaluation, which limits the generalizability of the findings. Similarly, in Table 5, the proposed method is evaluated exclusively on BLIP-2. These limitations weaken the empirical support for the effectiveness and robustness of the proposed approach.
3. Given the rapid progress in vision-language models, the authors are encouraged to evaluate their method on more recent VLP models. In Table 6, the performance gap between the baselines and the proposed method is relatively small, which limits the impact of the results. Incorporating additional models, such as LLaVA, could better demonstrate the effectiveness and generalizability of the proposed approach.
4. Based on the descriptions of Algorithms 2 and 3, the proposed method appears to be largely independent of the original CroPA approach. Therefore, it would be beneficial for the authors to evaluate their method in the context of other adversarial attack frameworks to better assess its generalizability and broader applicability.
5. The overall organization of the paper needs improvement. Algorithms 2 and 3 contain excessive empty space and should be reformatted to appear more compact and readable. Furthermore, the paper lacks a dedicated Related Work section. Thus, I strongly recommend that the authors undertake a thorough rewrite of the paper

---

> ### Author Response · Authors · 2025-05-01
> **Response to Reviewer PcXE (Part 1/4)**
>
> Thank you for your constructive feedback on our paper, we appreciate your effort to comprehensively evaluate the paper and credibility of our experiments.
>
> **W1: Reproducibility as a key contribution**
> > While reproducing previous work is important, it should not be positioned as a key contribution of the paper. Instead, the authors should focus on clearly identifying the limitations of the original CroPA method and emphasize how these shortcomings motivate the development of their proposed improvements.
>
> We do appreciate the your feedback and agree that identifying the limitations of the original CroPA method is crucial for motivating meaningful advancements, and that reproducibility shouldn’t be the only standalone objective. Aligning with these idea indeed, in our work, we treat reproduction not as a standalone contribution, but as a necessary and rigorous first step before any additional work, especially given that CroPA introduced a novel subfield of Cross-Prompt Attacks. Through this comprehensive reproduction, we uncovered several limitations in the original implementation, particularly regarding initialization susceptibility, cross-model and cross-image transferability (now updated to be more explicitly discussed in Sections 5.1, 5.2 and 5.3). These insights directly informed the design of our improvements.
>
> Our paper clearly outlines these shortcomings and presents extensive ablation studies that demonstrate how our enhancements address them, leading to more robust and generalizable adversarial attacks. We believe this progression, from verification to informed improvement, provides clarity and serves as a reliable baseline for future work. Our comprehensive reproducibility study represents one of our core contributions, with other core objectives being  the same principles  pointed out by the review.
>
> Additionally we have attempted to strengthen the algorithm, enhancing it’s cross-model transferability as well as inherent performance. It is in our understanding that this aligns with JMLR submission guidelines, promoting work centered towards reproducibility, analysis and verification of claims of previous works, and we hope our work to be a potential candidate for the Reproducibility Certification offered to credible papers oriented towards reproducibility.
>
> However, we recognize several much needed points in the feedback for clarification and expansion of the experiments and provide the relevant revisions with respect to each of them in the following parts.

---

> ### Author Response · Authors · 2025-05-01
> **Response to Reviewer PcXE (Part 2/4)**
>
> We have now updated the Additional Investigative Experiments sections and corresponding subsections to include clarifications stemming from limitations of CroPA and correspondingly proposing an enhancement to mitigate those limitations. The introductory subsections 5.1 , 5.2 and 5.3 are updated with this view in mind. The discussion section is also now organized to address the improvement of those limitations by our integrations, with a subsection now dedicated to each.
>
> The key limitations identified were :
>
> - The tendency of CroPA’s perturbations to be non transferable, remaining tightly coupled to the low-level features of the image in question and also to the architecture , to counteract these, we explore cross-image and cross-model transferability with our data augmentation and D-UAP enhancements. (Subsection - 5.2, 5.3, 7.1, 7.2 and 7.3 are now updated to reiterate this)
>
>   - The author’s too observe CroPA to be inherently non-transferable and overfitting to the model/image (we have presented their findings for reference in Appendix A.1)
>    - Our incorporated image augmentation techniques provide a notable increase in cross-image transferability, but we do also recognize (Discussion 7.3.1)  that further improving cross-image transferability in the context of cross-prompt attacks will require more sophisticated approaches beyond image augmentations
>   - CutMix as an image augmentation technique was also included in the cross-image analysis and it’s added advantages discussed in the method’s motivations (Subsection 5.3 and 5.3.2). Rationale about the empirical findings (Table 5 under subsection  6.5.2) have been discussed critically in 7.3.1. Potential trade-off between Cross Prompt and Cross Image transferability paradigms are also  now addressed in the updated discussion subsection 7.4.2
>   - Further, our investigation on cross-model transferability grounded in the advantages offered by D-UAP were updated with well-motivated. reasoning in subsection 5.1, empirical verification of the claims are now also updated to be more comprehensive across multiple models in sub-subsection 6.5.4 Table 8 (Previously Table 6), we have also provided further updated analysis of our own method’s limiting cases and what empirical verification of them suggests for future directions under sub-subsection 6.5.4
>
> - We hypothesized that CroPA’s bi-level optimization is susceptible to initialization changes, as is usually the case with bi-level optimizations on complex task-surfaces, 6.5.1 is now updated  theoretically verify this by extensive literature review and empirical evidence in sub-subsection 6.5.1 under the paragraph titled “Theoretically Analysis of Initialization in Bi-Level  Optimization”, and conclusive evidence was provided by our results under 6.5.1 with Table 4 and Figure 7,  and the results of our improved initialization depicted in Table 3, showing consistent gains over the CroPA and the baselines. The main identified points were :
>     1. CroPA's min-max formulation exhibits high sensitivity to initialization conditions, requiring carefully designed starting points for stable convergence across the saddle-point optimization landscape [1].
>     2. The asymmetric update schedule between image and prompt perturbations creates local cycling behaviors, making convergence heavily dependent on initial proximity to adversarially robust regions.
>     3. Semantic initialization provides superior cross-model transferability compared to random initialization, particularly effective in complex bi-level optimization contexts where pathways are less direct [2].
>
> [1] Zhang, J, Xiao, P, Sun, R & Luo, ZQ 2020, 'A single-loop smoothed gradient descent-ascent algorithm for nonconvex-concave min-max problems', Advances in Neural Information Processing Systems
>
> [2] Dongxian Wu, Shu-Tao Xia, and Yisen Wang. 2020. Adversarial weight perturbation helps robust generalization. In Proceedings of the 34th International Conference on Neural Information Processing Systems (NIPS '20)

---

> ### Author Response · Authors · 2025-05-01
> **Response to Reviewer PcXE (Part 3/4)**
>
> **W2: Insufficient experimental evaluation**
> >”The experimental evaluation of the proposed methods appears insufficient. For example, in Table 3, only three target prompts are used for evaluation, which limits the generalizability of the findings. Similarly, in Table 5, the proposed method is evaluated exclusively on BLIP-2. These limitations weaken the empirical support for the effectiveness and robustness of the proposed approach.”
>
> We are happy to report that additional empirical findings have been reported for the suggested sections of the paper :
>
> - Additional Model evaluations for DUAP method (previously present in Table-5) as well as an additional baseline comparison (CIA) along with PGD baseline (Multi-P)
> - Table-3 has now been updated to include  more diverse prompts and the additional CIA baseline, aiding to verify the updated initialization technique’s performance being better across the board
>
> **W3 and W4: Increased Evaluations on Models**
>
> >”Given the rapid progress in vision-language models, the authors are encouraged to evaluate their method on more recent VLP models. In Table 6, the performance gap between the baselines and the proposed method is relatively small, which limits the impact of the results. Incorporating additional models, such as LLaVA, could better demonstrate the effectiveness and generalizability of the proposed approach”
>
> >”Based on the descriptions of Algorithms 2 and 3, the proposed method appears to be largely independent of the original CroPA approach. Therefore, it would be beneficial for the authors to evaluate their method in the context of other adversarial attack frameworks to better assess its generalizability and broader applicability.”
>
> In light of the above feedback we have extensively updated our cross-model experiments to include LLaVA (Kindly refer Table 8), which was especially challenging since the codebase provided by the authors used a customized version of Transformers (v4.33.2), which did not support additional Vision-Language Models (VLMs) required for applying CroPA. Consequently, we had to extensively rework the entire codebase to integrate LLaVA for additional cross-model experiments.
>
> Kindly note that we mistakenly used different target texts while reporting the VQA results in Table 5, specifically, "unknown" for CroPA and Multi-P, and "bomb" for CroPA with DUAP. This was an unfortunate and rare oversight in our evaluation process. Upon discovering the inconsistency, we thoroughly reviewed and re-ran all relevant experiments to ensure consistency across all comparisons. We sincerely apologize for the earlier discrepancy. This error led to the impression of marginal gains in the transferability experiments which were not representative of the implemented method’s performance.
>
> Since CroPA pioneered the concept of Cross Prompt Transferability, there have been no credible benchmarks or established methods in this space since its introduction. However, through extensive research on related literature we did find one attack which aimed to put itself in comparison to the method proposed by CroPA. Context-Injection Attack (CIA) introduced by Yang et. (2024) positions itself to compare against CroPA in Cross-Prompt Transferability scenarios, and was hence perceived as a reasonable addition as a baseline to compare against in our results presented in Table 4 (6.5.3) and Table 6 & 7 (6.5.1), Multi-P and Single-P represent the PGD attack, which make up the standard baseline for CroPA.
>
> However, we excluded CIA when comparing transferability as upon experimentation CIA yielded negligible transferability, and presented the following limitations for transferability fundamentally in the method itself:
>
> - **Image-Specific Semantic Embedding**: CIA modifies the **visual token representation** of a particular image to embed a target concept (e.g., "dog") into it. However, the gradient direction and token space for a cat image versus a building image are vastly different. So, the same perturbation **won’t make sense** in another image’s latent representation—each image has its own visual-semantic embedding structure.
> - **Model-Specific Embedding Architectures** : CIA is highly dependent on the **structure of visual and textual encoders** (e.g., BLIP: A perturbation designed to affect position 10 of BLIP2 may not even correspond to a visual token in InstructBLIP or LLaVA).  Each model maps image features to different subspaces or token positions (e.g., BLIP2’s first 32 tokens are visual tokens).
>
> This reasoning is also updated in the Evaluation Rationale for our transferability experiments under sub-subsection 5.4.3.

---

> ### Author Response · Authors · 2025-05-01
> **Response to Reviewer PcXE (Part 4/4)**
>
> **W5: Organization and Structure**
> >”The overall organization of the paper needs improvement. Algorithms 2 and 3 contain excessive empty space and should be reformatted to appear more compact and readable. Furthermore, the paper lacks a dedicated Related Work section. Thus, I strongly recommend that the authors undertake a thorough rewrite of the paper”
>
> We recognize the missing features and organizational shortcomings, and appreciate the above feedback. With regard to the suggested changes we have implemented a Related Works section in which we comprehensively review related literature and  categorize them into distinct methodological streams. Additionally we have restructured Algorithms 2 and 3 (now moved to Appendix B for better organization of the paper). We have thoroughly revamped multiple parts of the paper (Discussion, Threat-Model Section, Analysis of results, Related Works and Evaluation Rationale) to make it more structurally appealing as well provide a clear logical flow to readers referring the paper, with every claim having the reasoning for the hypotheses and the corresponding empirical evidence.
>
> We have noted and implemented the requested changes to the best of our knowledge and ability, and we sincerely hope these changes address the concerns over the paper.

---

### Decision · Action_Editor_S33v · 2025-06-13

**Recommendation:** Accept as is

**Audience:**

Yes

**Audience Explanation:**

Needless to say, the interest in vision-language models is growing very rapidly in recent years. Their vulnerability to adversarial attacks deserves more attention in the research community. This paper will be of interest to researchers and practitioners in the TMLR community who are interested in the adversarial robustness of large vision-language models.

**Claims And Evidence:**

Yes

**Claims Explanation:**

This paper presents a comprehensive reproducibility study of a previous work on adversarial transferability in the context of large vision-language models. In addition, it proposes several improvements which are supported by extensive experiments. Although a number of issues were raised by the reviewers in their original reviews, they were mostly minor in nature and addressable. The authors have conducted further experiments and revised their original paper to address the comments. The paper in its current form is ready for publication in TMLR.

---

> ### Author Response · Authors · 2025-06-17
> **Response to Final Decision**
>
> Thank you very much for the acceptance, and for the thoughtful and constructive engagement throughout the review process.
>
> We’re especially grateful to our Action Editor for their support, responsiveness, and steady guidance at every stage. We also sincerely thank the reviewers for their detailed feedback and suggestions, which significantly improved the quality of the paper.
>
> This has been a genuinely collaborative and positive experience, and we’re very thankful to everyone involved. We will be sure to upload the camera-ready version promptly.
>
> Best regards,
> The Authors